# Constructing a Good Behavior Basis for Transfer using Generalized Policy Updates

**Safa Alver**
Mila, McGill University
safa.alver@mail.mcgill.ca

**Doina Precup**
Mila, McGill University and DeepMind
dprecup@cs.mcgill.ca

## Abstract

We study the problem of learning a good set of policies, so that when combined together, they can solve a wide variety of unseen reinforcement learning tasks with no or very little new data. Specifically, we consider the framework of generalized policy evaluation and improvement, in which the rewards for all tasks of interest are assumed to be expressible as a linear combination of a fixed set of features. We show theoretically that, under certain assumptions, having access to a specific set of diverse policies, which we call a set of independent policies, can allow for instantaneously achieving high-level performance on all possible downstream tasks which are typically more complex than the ones on which the agent was trained. Based on this theoretical analysis, we propose a simple algorithm that iteratively constructs this set of policies. In addition to empirically validating our theoretical results, we compare our approach with recently proposed diverse policy set construction methods and show that, while others fail, our approach is able to build a behavior basis that enables instantaneous transfer to all possible downstream tasks. We also show empirically that having access to a set of independent policies can better bootstrap the learning process on downstream tasks where the new reward function cannot be described as a linear combination of the features. Finally, we demonstrate how this policy set can be useful in a lifelong reinforcement learning setting.

## 1 Introduction

Reinforcement learning (RL) studies the problem of building rational decision-making agents that maximize long term cumulative reward through trial-and-error interaction with a given environment. In recent years, RL algorithms combined with powerful function approximators such as deep neural networks have achieved significant successes in a wide range of challenging domains (see e.g. Mnih et al. (2015); Vinyals et al. (2019); Silver et al. (2017; 2018)). However, these algorithms require substantial amounts of data for performing very narrowly-defined tasks. In addition to being data-hungry, they are also very brittle to changes in the environment, such as changes in the tasks over time. The most important reasons behind these two shortcomings is that RL algorithms usually learn to perform a task from scratch, without leveraging any form of prior knowledge, and they are trained to optimize performance on only a single task.

A promising approach to tackle both of these shortcomings is to learn multiple ways of behaving, i.e., multiple policies that optimize different reward functions, and to reuse them as needed. Having access to multiple pre-learned policies can allow an agent to quickly solve reoccurring tasks in a lifelong RL setting. It can also allow for learning to combine the existing policies via a meta-policy to quickly learn new tasks, as in hierarchical RL. Recently, Barreto et al. (2020) have proposed the "generalized policy updates" framework, which generalizes the classical policy evaluation and policy improvement operations that underlie many of today's RL algorithms. Its goal is to allow reusing policies resulting from previously learned tasks in order to perform well on downstream tasks, while also being data-efficient. More precisely, after learning the successor features of several policies in a policy set, also referred to as a behavior basis, they were able to instantaneously "synthesize", in a zero-shot manner, new policies to solve downstream tasks, via generalized policy improvement. However, this work leaves open two important questions: (i) what set of policies should the agent

learn so that its instantaneous performance on *all* possible downstream tasks is guaranteed to be good, and (ii) under what conditions does such a set of policies exist.

In this paper, we provide answers to the questions above by proving that under certain assumptions about the environment dynamics and features, learning a diverse set of policies, which we call a *set of independent policies*, indeed guarantees good instantaneous performance on all possible downstream tasks. After providing an iterative algorithm for the construction of this set, we perform several experiments that validate our theoretical findings. In addition to the validation experiments, we compare this algorithm with recently proposed diverse policy set construction methods (Eysenbach et al., 2018; Zahavy et al., 2020; 2021) and show that, unlike these methods, our approach is able to construct a behavior basis that enables instantaneous transfer to all possible tasks. We also show empirically that learning a set of independent policies can better bootstrap the learning process on downstream tasks where the reward function cannot be described by a linear combination of the features. Finally, we demonstrate the usefulness of this set in a lifelong RL scenario, in which the agent faces different tasks over its lifetime. We hope that our study will bring the community a step closer to building lifelong RL agents that are able to perform multiple tasks and are able to instantaneously/quickly adapt to new ones during its lifetime.

## 2 BACKGROUND

**Reinforcement Learning.** In RL (Sutton & Barto, 2018), an agent interacts with its environment by choosing actions to get as much as cumulative long-term reward. The interaction between the agent and its environment is usually modeled as a Markov Decision Process (MDP). An MDP is a tuple $M \equiv (\mathcal{S}, \mathcal{A}, P, r, d_0, \gamma)$, where $\mathcal{S}$ is the (finite) set of states, $\mathcal{A}$ is the (finite) set of actions, $P : \mathcal{S} \times \mathcal{A} \times \mathcal{S} \to [0, 1]$ is the transition distribution, $r : \mathcal{S} \times \mathcal{A} \times \mathcal{S} \to \mathbb{R}$ is the reward function, which specifies the task of interest, $d_0 : \mathcal{S} \to [0, 1]$ is the initial state distribution and $\gamma \in [0, 1)$ is the discount factor. In RL, typically the agent does not have any knowledge about $P$ and $r$ beforehand, and its goal is to find, through pure interaction, a policy $\pi : \mathcal{S} \to \mathcal{A}$ that maximizes the expected sum of discounted rewards $E_{\pi,P}[\sum_{t=0}^{\infty} \gamma^t r(S_t, A_t, S_{t+1}) | S_0 \sim d_0]$, where $E_{\pi,P}[\cdot]$ denotes the expectation over trajectories induced by $\pi$ and $P$.

**Successor Features.** The successor representation for a state $s$ under a policy $\pi$ allows $s$ to be represented by the (discounted) distribution of states encountered when following $\pi$ from $s$ (Dayan, 1993). Given a policy, successor features (SF, Barreto et al., 2017) are a generalization of the idea of successor representations from the tabular setting to function approximation. Following Barreto et al. (2017), we define SFs of a policy $\pi$ for state-action $(s, a)$ as:

$$\boldsymbol{\psi}^\pi(s, a) \equiv E_{\pi,P}\left[\sum_{i=0}^{\infty} \gamma^i \boldsymbol{\phi}(S_{t+i}, A_{t+i}, S_{t+i+1}) \Big| S_t = s, A_t = a\right], \tag{1}$$

where the $i$th component of $\boldsymbol{\psi}^\pi$ gives the expected discounted sum of the $i$th component of the feature vector, $\phi_i$, when following policy $\pi$, starting from the state-action pair $(s, a)$.

Successor features allow a decoupling between the reward function and the environment dynamics. More concretely, if the reward function for a task can be represented as a linear combination of a feature vector $\boldsymbol{\phi}(s, a, s') \in \mathbb{R}^n$:

$$r_{\mathbf{w}}(s, a, s') = \boldsymbol{\phi}(s, a, s')^\top \cdot \mathbf{w}, \tag{2}$$

where $\mathbf{w} \in \mathbb{R}^n$, then, as we will detail below, the state-action value function $Q_{r_{\mathbf{w}}}^\pi$ can be computed immediately as the dot-product of $\boldsymbol{\psi}^\pi$ and $\mathbf{w}$. The elements of $\mathbf{w}$, $w_i$, can be viewed as indicating a "preference" towards each of the features. Thus, we refer to $\mathbf{w}$ interchangeably as either the preference vector or the task. Intuitively, the elements $\phi_i$ of the feature vector $\boldsymbol{\phi}$ can be viewed as salient events that maybe desirable or undesirable to the agent, such as picking up or leaving objects of certain type, and reaching and/or avoiding certain states.

**Generalized Policy Evaluation and Improvement.** Generalized Policy Evaluation (GPE) and Generalized Policy Improvement (GPI), together referred to as Generalized Policy Updates, are generalizations of the well-known policy evaluation and policy improvement operations in standard dynamic programming to a set of tasks and a set of policies (Barreto et al., 2020). They are used as a transfer mechanism in RL to quickly construct a solution for a newly given task. One particularly

efficient instantiation of GPE & GPI is through the use of SFs and value-based action selection. More concretely, given a set of MDPs having the following form:

$$\mathcal{M}_\phi \equiv \{(\mathcal{S}, \mathcal{A}, P, r_\mathbf{w} = \boldsymbol{\phi} \cdot \mathbf{w}, d_0, \gamma) | \mathbf{w} \in \mathbb{R}^n\}, \tag{3}$$

and given SFs $\boldsymbol{\psi}^\pi(s, a)$ of a policy $\pi$, an efficient form of GPE on the task $r_\mathbf{w}$ for policy $\pi$ can be performed as follows:

$$\boldsymbol{\psi}^\pi(s, a)^\top \cdot \mathbf{w} = Q_{r_\mathbf{w}}^\pi(s, a), \tag{4}$$

where $Q_{r_\mathbf{w}}^\pi(s, a)$ is the state-action value function of $\pi$ on the task $r_\mathbf{w}$. And, after performing GPE for all the policies $\pi$ in a finite policy set $\Pi$, following Barreto et al. (2017), an efficient form of GPI can be performed as follows:

$$\pi_\Pi^{\text{GPI}}(s) \in \arg\max_{a \in \mathcal{A}} \max_{\pi \in \Pi} Q_{r_\mathbf{w}}^\pi(s, a). \tag{5}$$

We will refer to this specific use of SFs and value-based action selection for performing GPE & GPI as simply GPE & GPI throughout the rest of this study. Note that $\pi^{\text{GPI}}$ will in general outperform all the policies in $\Pi$, and that the actions selected by $\pi^{\text{GPI}}$ on a state may not coincide with any of the actions selected by $\pi \in \Pi$ on that state. Hence, the policy space that can be attained by GPI can in principle be a lot larger than, e.g., the space accessible by calling policies sequentially from the original set.

## 3 PROBLEM FORMULATION AND THEORETICAL ANALYSIS

GPE & GPI provide a guarantee that, for any reward function linear in the features, $\pi^{\text{GPI}}$ is at least as good as any of the policies $\pi$ from the "base set" which was used to construct it. While this is an appealing guarantee of monotonic improvement, it does not say much, for two reasons. First, it is not clear how big an improvement can be expected for different tasks. More importantly, it leaves open the question of how one should choose base policies in order to ensure as much improvement as possible. After all, if we had a weak set of policies and we simply matched their value with $\pi^{\text{GPI}}$, this may not be very useful. We will now show that, under certain assumptions, having access to a specific set of diverse policies, which we call a set of independent policies, can allow for instantaneously achieving high-level performance on *all* possible downstream tasks.

Let us start by assuming that we are interested in a set of MDPs $\mathcal{M}_\phi$, as defined in (3), with deterministic transition functions (the reason for the determinism assumption will become clear by the end of this section). For convenience, we also restrict the possible $\mathbf{w}$ values from $\mathbb{R}^n$ to $\mathcal{W}$, where $\mathcal{W}$ is the surface of the $\ell_2$ $n$-dimensional ball. Note that this choice does not alter the optimal policies of the MDPs in $\mathcal{M}_\phi$, as an optimal policy is invariant with respect to the scale of the reward and $\mathcal{W}$ contains all possible directions. Next, we assume that the features $\phi_i$ that make up the feature vectors $\boldsymbol{\phi}$ form a *set of independent features (SIF)*, defined as follows:

**Definition 1** (SIF). *A set of features $\Phi = \{\phi_i | \phi_i : \mathcal{S} \times \mathcal{A} \times \mathcal{S} \to \{0, C\}, C \in \mathbb{R}_+\}_{i=1}^n$ is called* independent *if, for any feature $\phi_i \in \Phi$ and any initial state $s_0 \sim d_0$, we have: (i) $\phi_i(s_0, a_0, s_1) = 0$ $\forall a_0 \in \mathcal{A}$ and $\forall s_1 \sim P(s_0, a_0, \cdot)$, and (ii) there exists at least one trajectory, starting from $s_0$, in which all the states associated with $\phi_i(s_t, a_t, s_{t+1}) = C$ are visited, while the states associated with $\phi_j(s_t, a_t, s_{t+1}) = C$, $\forall j \neq i$, are not visited.*

It should be noted that a specific instantiation of this definition is the case where each feature is set to a positive constant at certain independently reachable state/states and zero elsewhere, which is the most common instantiation of feature vectors used in previous related work (Barreto et al., 2017; 2020).

We define the performance of an arbitrary policy $\pi$ on a task $r_\mathbf{w}$ as:

$$J_{r_\mathbf{w}}^\pi \equiv E_{\pi, P}\left[\sum_{t=0}^\infty r_\mathbf{w}(S_t, A_t, S_{t+1}) \Big| S_0 \sim d_0\right], \tag{6}$$

where $E_{\pi, P}[\cdot]$ denotes the expectation over trajectories induced by $\pi$ and $P$. Note that $J_{r_\mathbf{w}}^\pi$ is a scalar corresponding to the expected undiscounted return of policy $\pi$ under the initial state distribution $d_0$, which is the expected total reward obtained by $\pi$ when starting from $s_0 \sim d_0$. We are now ready to formalize the problem we want to tackle:

**Problem Formulation.** Given a set of MDPs $\mathcal{M}_\phi$ with deterministic transition functions and a SIF, we want to construct a set of $n$ policies $\Pi^n = \{\pi_i\}_{i=1}^n$, such that the performance of the policy $\pi_{\Pi^n}^{\text{GPI}}$ obtained by performing GPI on that set will maximize (6) for all rewards $r_{\mathbf{w}}$, where $\mathbf{w} \in \mathcal{W}$. That is, we want to solve the following problem:

$$\arg \max_{\Pi^n \subseteq \Pi} J_{r_{\mathbf{w}}}^{\pi_{\Pi^n}^{\text{GPI}}} \text{ for all } \mathbf{w} \in \mathcal{W}. \tag{7}$$

It should be noted that the performance measure provided in (6) only measures the expected total reward and thus cannot capture the optimality of the GPI policy. For instance, this measure cannot distinguish between two policies that achieve the same expected total reward in a different number of time steps. However, Theorem 2 in Barreto et al. (2017) implies that, in general, the only way to guarantee the optimality of the GPI policy is to construct a behavior basis that contains all possible policies induced by all $\mathbf{w} \in \mathcal{W}$. Since there are infinitely many $\mathbf{w}$ values, this is impractical. Thus, in this study, we only consider GPI policies that maximize the expected total reward (6).

As a solution candidate to the problem in (7), we now focus on a specific set of deterministic policies, called *set of independent policies (SIP)*, that are able to obtain features independently of each other:

**Definition 2** (SIP). *Let $\Phi = \{\phi_i\}_{i=1}^n$ be a SIF and let $\Pi = \{\pi_i\}_{i=1}^n$ be a set of deterministic policies that are induced by each of the features in $\Phi$. $\Pi$ is defined to be a SIP if its elements, $\pi_i$, satisfy:*

$$\phi_j(s_t, a_t, s_{t+1}) = \phi_j(s_0, a_0, s_1) \quad \forall j \neq i, \forall i, \forall s_0 \sim d_0 \text{ and } \forall t \in \{1, \dots, T\}, \tag{8}$$

*where $T$ is the horizon in episodic environments and $T \to \infty$ in non-episodic ones, $a_0 = \pi_i(s_0)$ and $(s_t, a_t, s_{t+1})_{t=1}^T$ is the sequence of state-action-state triples generated by $\pi_i$'s interaction with the environment.*

In general, having a SIP in a set of MDPs with stochastic transition functions is not possible, as the stochasticity can prevent (8) from holding. Thus, the assumption of a set of MDPs with deterministic transition functions is critical to our analysis. An immediate consequence of having this set of policies is that the corresponding SFs can be expressed in a simpler form, as follows:

**Lemma 1.** *Let $\Phi$ be a SIF and let $\pi_i$ be a policy that is induced by the feature $\phi_i \in \Phi$ and is a member of a SIP $\Pi$. Then, the entries of the SF $\boldsymbol{\psi}^{\pi_i}$ of policy $\pi_i$ has the following form:*

$$\psi_j^{\pi_i}(s, a) = \begin{cases} \psi_i^{\pi_i}(s, a), & \text{if } i = j \\ 0, & \text{otherwise} \end{cases}. \tag{9}$$

Due to the space constraints, we provide all the proofs in the supp. material. Lemma 1 implies that once we have a SIP, the SFs take the much simpler form (9), which can allow for the GPI policy to maximize performance on *all* possible tasks according to the measure in (6), solving the optimization objective in (7):

**Theorem 1.** *Let $\Phi$ be a SIF and let $\Pi$ be a SIP induced by each of the features in $\Phi$. Then, the GPI policy $\pi_\Pi^{\text{GPI}}$ is a solution to the optimization problem defined in (7).*

Theorem 1 implies that having access to a SIP which consists of only $n$ policies, where $n$ is the dimensionality of $\phi$, and applying the policy composition operator GPI on top, allows for instantaneously achieving maximum performance across all possible downstream tasks. Considering the fact that there are *infinitely many* such tasks, this provides a significant gain in the number of policies that are required to be learned for full downstream task coverage.

## 4 CONSTRUCTING A SET OF INDEPENDENT POLICIES

Based on our theoretical analysis in Section 3, we now propose a simple algorithm that iteratively constructs a SIP, given a SIF. Since independent policies can attain certain features without affecting the others, a set of them can be constructed by simply learning policies for each of the appropriate tasks and then by adding these learned policies one-by-one to an initially empty set. In particular, by sequentially learning policies for each of the tasks in $W = \{\mathbf{w}_i\}_{i=1}^n \subset \mathcal{W}$, where $\mathbf{w}_i$ is an $n$-dimensional vector with a positive number in the $i$th coordinate and negative numbers elsewhere, and then adding these policies to a policy set, one can construct a SIP. Algorithm 1 provides a step-by-step description of this construction process. Note that the algorithm does not depend on the

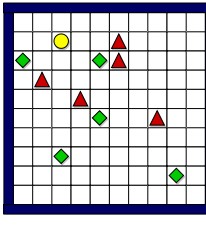

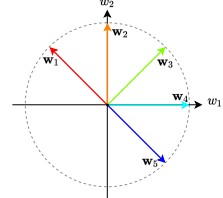

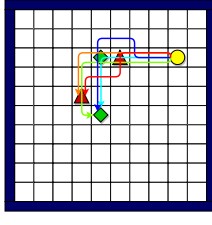

(a) Environment          (b) Preference vectors          (c) Trajectories

Figure 1: (a) The 2D item collection environment (Barreto et al., 2020). The environment consists of $10 \times 10$ cells and the shape (traingle and diamond) of the items represents their type. (b) Five distinct preference vectors that lie on the surface of the $\ell_2$ 2D ball: $\mathbf{w}_1 = (-\sqrt{1/2} \ +\sqrt{1/2})$, $\mathbf{w}_2 = (0 \ +1)$, $\mathbf{w}_3 = (+\sqrt{1/2} \ +\sqrt{1/2})$, $\mathbf{w}_4 = (+1 \ 0)$ and $\mathbf{w}_5 = (+\sqrt{1/2} \ -\sqrt{1/2})$. (c) The trajectories taken by the optimal policies ($\gamma = 0.95$) corresponding to the preference vectors $\mathbf{w}_1, \ldots, \mathbf{w}_5$ in a simplified version of the environment with two items of each type.

particular values of $\mathbf{w}_i \in W$ as long as the corresponding coordinate has a positive value and all others are negative. Another thing to note is that the algorithm runs for only $n$ iterations, where $n$ is the dimensionality of the feature vector $\boldsymbol{\phi}$.

## 5 EXPERIMENTS

We start this section by performing experiments to illustrate the theoretical results derived in Section 3. Then, we compare the performance of the policy construction approach presented in Algorithm 1 with recently proposed diverse policy set construction methods (Eysenbach et al., 2018; Zahavy et al., 2020; 2021). Next, we show that a SIP can better bootstrap learning on new tasks where the reward function cannot in fact be expressed as a linear combination of the features, a case that lies outside of our assumptions. Finally, we demonstrate

---

**Algorithm 1:** Constructing a SIP

**Require:** a SIF $\Phi$ ;
**Initialize:** $\Pi^0 \leftarrow \{\}, t \leftarrow 1, W \leftarrow \{\mathbf{w}_i\}_{i=1}^n$ ;
**while** $t \leq n$ **do**
     $\mathbf{w}_t \leftarrow W[t]$ ;
     $\pi^t \leftarrow$ solution of the task $r_{\mathbf{w}_t}$ using RL ;
     $\Pi^t \leftarrow \Pi^{t-1} + \{\pi^t\}$ ;
     $t \leftarrow t + 1$ ;
**end**
**return:** $\Pi^n$

---

that the set of policies produced by our approach can be useful in a lifelong RL setting. The experimental details together with more detailed results can be found in the supp. material.

**Experimental Setup.** Throughout this section, we perform experiments on the 2D item collection environment proposed in Barreto et al. (2020) (see Fig. 1a), as it is a *prototypical* environment where GPE & GPI is useful and it allows for easy visualization of performance across all downstream tasks. Here, the agent, depicted by the yellow circle, starts randomly in one of the cells and has to obtain/avoid 5 randomly placed red and green items, which are of different type. At each time step, the agent receives an image that contains its own position and the position and type of each item. Based on this, the agent selects an action that deterministically moves it to one of its four neighboring cells (except if the agent is adjacent to the boundaries of the grid, it remains on the same cell). By moving to the cells occupied by an item, the agent picks up that item and gets a reward defined by that item type. The goal of the agent is to pick up the "good" items and avoid the "bad" ones, depending on the preference vector $\mathbf{w}$. The agent-environment interaction lasts for 50 time steps, after which the agent receives a "done" signal, marking the end of the episode. Note that despite its small size, the cardinality of this environment's state space is on the order of $10^{15}$ and thus requires the use of function approximation. More on the implementation details of the environment and results on its stochastic version can be found in the supp. material.

In order to use GPE & GPI in this environment, we must first define: (i) the features $\boldsymbol{\phi}$ and (ii) a set of policies $\Pi$. Following Barreto et al. (2020), we define each feature $\phi_i$ as an indicator function signalling whether an item of type $i$ has been picked up by the agent. That is, $\phi_i(s, a, s') = 1$ if taking action $a$ in state $s$ results in picking up an item of type $i$, and $\phi_i(s, a, s') = 0$ otherwise. Note that these form a SIF, as there exist trajectories which start as $\boldsymbol{\phi}(s_0, a_0, s_1) = 0$ and in which all the features associated with a certain item type can be obtained without affecting the others. We now

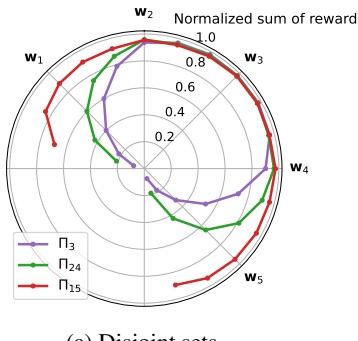 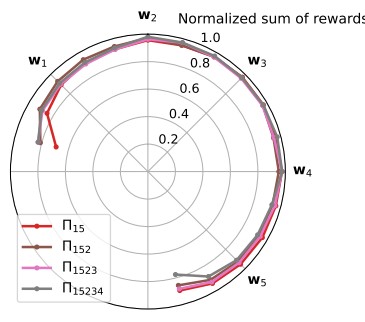

(a) Disjoint sets            (b) Incrementally growing sets

Figure 2: The normalized sum of rewards over 17 evenly spread tasks over the nonnegative quadrants of the unit circle. The plots are obtained by averaging over 10 runs with 1000 episodes for each task. The performance comparison of $\Pi_{15}$ (a) with disjoint sets $\Pi_{24}$ and $\Pi_5$, and (b) with incrementally growing sets $\Pi_{152}$, $\Pi_{1523}$ and $\Pi_{15234}$.

turn to the question on how to determine the set of policies $\Pi$. We restrict the policies in $\Pi$ to be solutions of the tasks $\mathbf{w} \in \mathcal{W}$. With Algorithm 1, we have already provided one way of constructing this set. Throughout the rest of this section we will compare Algorithm 1 with alternative policy set construction methods.

Following Barreto et al. (2020), we use an algorithm analogous to Q-learning to approximate the SFs induced by $\phi$ and $\Pi$. Similarly, we represent the SFs using multilayer perceptrons with two hidden layers. More details can be found in Barreto et al. (2020).

## 5.1 ILLUSTRATIVE EXPERIMENTS

**Question 1.** *Do the theoretical results, derived in Section 3, also hold empirically?*

In order to answer this question, we constructed a policy set using Algorithm 1 and evaluated it using the evaluation scheme proposed by Barreto et al. (2020). Specifically, after initializing $W = \{\mathbf{w}_1, \mathbf{w}_5\}$ and running Algorithm 1, we obtained the SIP $\Pi_{15} = \{\pi_1, \pi_5\}$, whose elements are solutions to the tasks $\mathbf{w}_1$ and $\mathbf{w}_5$, respectively (see Fig. 1b). Then, we ran an evaluation scheme that evaluates the GPI policy $\pi_{\Pi_{15}}^{\text{GPI}}$, over 17 evenly spread tasks over the nonnegative quadrants of the unit circle in Fig. 1b. We did not perform evaluations in the negative quadrant, as the tasks there are not interesting. Results are shown in Fig. 2a. As can be seen, $\pi_{\Pi_{15}}^{\text{GPI}}$ performs well across all the downstream tasks, empirically verifying Theorem 1.

**Question 2.** *Is having a SIP essential for full downstream task coverage?*

According to Theorem 1, having a SIP is a requirement for guaranteeing good performance across all downstream tasks. However, in order to see how well the GPI policy performs when this is not the case, similar to Barreto et al. (2020), we compared $\Pi_{15}$ with two other possible policy sets: $\Pi_{24} = \{\pi_2, \pi_4\}$ and $\Pi_3 = \{\pi_3\}$, whose policies correspond to the tasks $\mathbf{w}_2$, $\mathbf{w}_4$ and $\mathbf{w}_3$, respectively. It should be noted that $\Pi_{24}$ is a default policy set used in prior studies on SFs (Barreto et al., 2017; 2018; 2020; Borsa et al., 2018). The results, in Fig. 2a, show that $\pi_{\Pi_{24}}^{\text{GPI}}$ and $\pi_{\Pi_3}^{\text{GPI}}$ are not able to perform well across all downstream tasks, specifically failing on quadrants II and IV. This justifies the requirement of learning a SIP for full downstream task coverage.

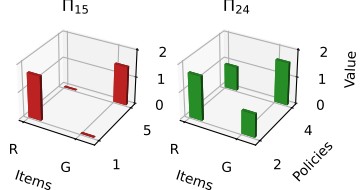

Figure 3: The learned SFs of the policies in $\Pi_{15}$ and $\Pi_{24}$ for the state in Fig. 1c and the left action. Here, each row corresponds to a different policy and each column corresponds to a different item ('R'ed and 'G'reen).

**Question 3.** *Do the SFs of the policies in a SIP have a simple form?*

According to Lemma 1, the SFs of the policies in a SIP should have a simple form. To empirically verify this, in Fig. 3, we visualize the SFs of the policies in the sets $\Pi_{15}$ and $\Pi_{24}$ for the simplified environment depicted in Fig. 1c. We can see that the SFs of the policies in $\Pi_{15}$ indeed have a simple

form, in which the off-diagonal entries have values that are very close to zero, while this is not the case for $\Pi_{24}$.

**Question 4.** *Does adding more policies to a SIP have any effect on its downstream task coverage?*

Theorem 1 implies that, given a SIF, having a SIP is enough to guarantee good performance across all possible tasks. To test empirically whether adding more policies to a SIP has any effect on its performance across all tasks, we compare the performance of the policy set $\Pi_{15}$ with policy sets $\Pi_{152}$, $\Pi_{1523}$ and $\Pi_{15234}$, formed by adding one-by-one the policies $\pi_2$, $\pi_3$ and $\pi_4$ to $\Pi_{15}$. Results are shown in Fig. 2b. As expected, adding more policies to the SIP $\Pi_{15}$ has no effect on its downstream task coverage.

## 5.2 COMPARATIVE AND LIFELONG LEARNING EXPERIMENTS

**Question 5.** *How does Algorithm 1 compare to prior diverse policy set construction methods?*

Prior work has shown that having access to a diverse set of policies can help transfer to downstream tasks. Algorithm 1 can also be seen as a diverse policy set construction method, as it constructs a policy set that is diverse in the feature visitation profile. In order to see how well it compares to prior work, we compare Algorithm 1 with three recently proposed diverse policy set construction methods: (i) DIAYN[1] (Eysenbach et al., 2018) which constructs a policy set by collectively training policies with information-theoretic reward functions that encourage discriminablity among the policies in the set based on their state visitation profile, (ii) SMP (Zahavy et al., 2020) which constructs a policy set by iteratively adding policies trained on the worst-case linear reward with respect to the previous policies in the set, and (iii) DSP (Zahavy et al., 2021) which constructs a policy set by iteratively adding policies that are trained on tasks $\mathbf{w}_n = -(1/n)\sum_{k=0}^{n}\boldsymbol{\psi}_k$, where $n$ is the number of policies in the current policy set. More on the specific implementation details of these methods can be found in the supp. material. The comparison results, shown in Fig. 4, indicate that none of the prior diverse policy set construction methods are able to consistently construct a policy set that enables full downstream task coverage.

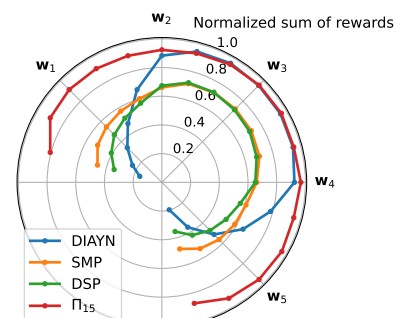

Figure 4: The normalized sum of rewards of $\Pi_{15}$, and the policy sets constructed by DIAYN, SMP and DSP. Since the policy sets constructed by the prior methods depend on their particular initialization, their plots are obtained by running each of the constructed policy sets for 5 runs and then averaging over their results. For each task, the agent was evaluated on 1000 episodes. The plot for $\Pi_{15}$ is obtained in a similar way as in Fig. 2.

**Question 6.** *Is learning a SIP also effective when the downstream task's reward function cannot be expressed a single linear combination of the features?*

So far, we have considered the scenario where the GPI policy is evaluated on tasks whose reward functions were obtained by linearly combining the features $\phi$ using a single, fixed $\mathbf{w}$. We now consider downstream tasks that do not satisfy this assumption, e.g. where $\mathbf{w}$ changes as a function over the environment's state. Note that Theorem 1 has nothing to say in this case. Nevertheless, in order to test whether if learning a SIP can still be useful in this scenario, we consider a transfer setting in which a meta-policy is trained to orchestrate the policies in this set when faced with a downstream task. Specifically, after constructing the policy set, we learn a function that maps states to preference vectors, $\omega : \mathcal{S} \to \mathcal{W}'$, whose output is then used by GPE & GPI to synthesize a policy for the downstream task of interest (see the "Preferences as Actions" section in Barreto et al. (2020) and the Option Keyboard framework (Barreto et al., 2019) for more details). We define $\mathcal{W}' = \{\mathbf{w}_1, \mathbf{w}_2, \mathbf{w}_3, \mathbf{w}_4, \mathbf{w}_5\}$ and use Q-learning to learn $\omega$.

As an instantiation of the scenario discussed above, we consider two different downstream tasks: (i) sequential reward collection, in which the agent must first pick up items of a certain type and then collect the items of the other type, and (ii) balanced reward collection, in which the agent must pick up the type of item that is more abundant in the environment. We compared the performance

---

[1]We use a GPE & GPI compatible version of DIAYN whose details are provided in the supp. material.

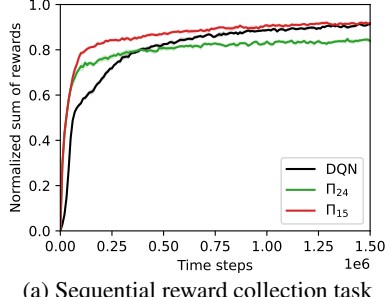 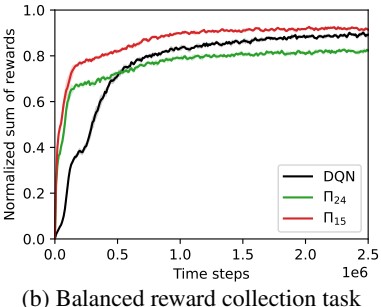

(a) Sequential reward collection task      (b) Balanced reward collection task

Figure 5: The normalized sum of rewards of the policy sets $\Pi_{15}$ and $\Pi_{24}$, and DQN on the (a) sequential reward collection and (b) balanced reward collection tasks. Shadowed regions are one standard error over 10 runs.

of the SIP $\Pi_{15}$ with the policy set $\Pi_{24}$. As a reference to the maximum reachable performance, we also provided the learning curve of DQN (Mnih et al., 2015), whose specific implementation details can be found in the supp. material. The results, shown in Fig. 5a and 5b, show that $\Pi_{15}$ outperforms $\Pi_{24}$ both in terms of the learning speed and asymptotic performance, reaching the asymptotic performance of DQN. These results suggest that having access to a SIP can also be effective in scenarios where the downstream task's reward function cannot be described as a single linear combination of the features.

**Question 7.** *How can learning a SIP be useful in a lifelong RL setting?*

Except for the last question, we have considered an idealistic scenario in which, during transfer, the agent was provided with a single preference vector $\mathbf{w}$ describing a stationary downstream task of interest. We now consider a lifelong RL (LRL) setting, in which the agent has to infer the current $\mathbf{w}$ from data and has to quickly adapt to the changing $\mathbf{w}$'s during its lifetime.[2] Concretely, we consider the setting in which the agent infers $\mathbf{w}$ by following the regression procedure provided in Barreto et al. (2020) and the tasks change from $\mathbf{w}_1$ to $\mathbf{w}_5$ one-by-one in a clockwise fashion every $5 \times 10^5$ time steps. After the end of task $\mathbf{w}_5$, the task resets back to $\mathbf{w}_1$ and this task cycle goes on forever.

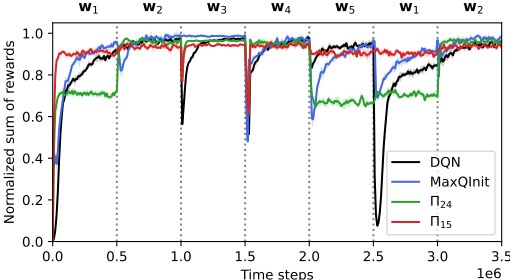

Figure 6: The normalized sum of rewards of the policy sets $\Pi_{15}$ and $\Pi_{24}$, DQN, and MaxQInit in a lifelong RL setting described in the text. Shadowed regions are one standard error over 100 runs.

In Fig. 6, we compare the performance of $\Pi_{15}$ with the policy set $\Pi_{24}$, with DQN, and with a LRL method MaxQInit (Abel et al., 2018). For fair comparison, we allow for MaxQInit to have access to all of the tasks before each task change (see the supp. material for the implementation details). We see that the policy set $\Pi_{15}$ allows for better transfer to changing tasks compared to $\Pi_{24}$. We also see that, although MaxQInit outperforms DQN, they both fail in instantaneously adapting to the changing tasks, and DQN sometimes catastrophically fails on tasks that it has already encountered before (see the drastic drop in performance when the task transitions from $\mathbf{w}_5$ to $\mathbf{w}_1$). These results illustrate how a SIP can also be useful in a LRL scenario.

## 6   RELATED WORK

**Successor Features and GPI.** In this study, we focused on relevant work that uses a set of SFs together with GPI for performing transfer in RL (Barreto et al., 2017; 2018; 2019; 2020; Borsa et al., 2018; Hansen et al., 2019). Among these studies, the study of Barreto et al. (2020) is the closest to our work. We built on top of their approach and tried to answer the following questions: (i) what behavior basis should be learned by the agent so that its instantaneous performance on

---

[2]We consider a setting in which the agent is allowed for a certain amount of pre-training before LRL starts.

all possible downstream tasks is guaranteed to be good, and (ii) what are the conditions required for this? In addition, we provided empirical results illustrating how a good behavior basis can be useful in more realistic scenarios. Another closely related work is the study of Grimm et al. (2019) which shows that having access to a set of policies that obtain certain disentangled features can allow for achieving high level performance on exponentially many downstream goal-reaching tasks. However, this study requires having access to disentangled features, which have a very specific form; in contrast, we require a set of independent features, which are more general and commonly used (Barreto et al., 2017; 2020). Additionally, we provide a more formal treatment of the instantaneous transfer problem and our experiments address a broader range of questions.

**Diverse Skill Discovery.** Another related line of research is the information theoretic diverse skill discovery literature (Gregor et al., 2016; Eysenbach et al., 2018; Achiam et al., 2018), in which information theoretic objectives are used to construct a diverse policy set that enables better exploration and transfer on downstream tasks. An important problem with learning these skills is that most of the discovered policies end up being uninteresting, in the sense that they do not come to be useful in solving downstream tasks of interest, filling up the set with not so interesting policies. Our work prevents this by constructing a policy set in which every policy is trained to achieve only a particular task so that when combined, can solve more complex downstream tasks of interest.

Other related literature includes the use of SFs to build a diverse skill set. Through using the criterion of robustness, Zahavy et al. (2020) proposes a method that constructs a set of policies that do as well as possible in the worst-case scenario, and shows that this set naturally becomes diverse. However, a problem with this policy construction method is that it is sensitive to the initialization of the preference vector and thus leads to policy sets that do not have full downstream task coverage. By using explicit diversity rewards, Zahavy et al. (2021) proposes another method that aims to minimize the correlation between the SFs of the policies in the constructed set. However, since this method depends on the evolution of the policy set, it also fails in building a policy set with full downstream tasks coverage; by contrast, Algorithm 1 does exactly this.

**Policy Reuse for Transfer and Lifelong RL.** Although there have been prior studies on policy reuse for faster learning on downstream tasks (Pickett & Barto, 2002; Fernández & Veloso, 2006; Machado et al., 2018; Barreto et al., 2018) and for achieving lifelong RL (Abel et al., 2018), none of them tackle the problem of learning a set of policies for full downstream task coverage. Recently, by building on top of Barreto et al. (2017), Nemecek & Parr (2021) proposes a policy set construction method that starts with a set of policies and gradually adds new ones based on a pre-defined threshold. However, they also do not consider the full downstream task coverage problem. The most closely related work however are the recent studies of Tasse et al. that improve on Van Niekerk et al. (2019) and in which a set of pre-learned base policies are logically composed for super-exponential downstream task coverage (Tasse et al., 2020) and lifelong RL (Tasse et al., 2022). Despite the similarities in the motivation, there are important differences compared to our work: (i) while their approach requires extensions to the reward and value function definitions, our approach builds on top of the readily available GPE & GPI framework and (ii) while their approach only considers goal-based tasks, where upon reaching a goal the episode terminates, our approach handles both these tasks and the ones in which the agent has to achieve multiple goals within a single fixed-length episode.

# 7 CONCLUSION AND FUTURE WORK

To summarize, in this study, we provided a theoretical analysis elucidating what counts as a good behavior basis for performing GPE & GPI and showed that, under certain assumptions, having access to a set of independent policies allows for instantaneously achieving high level performance on all downstream tasks. Based on this analysis, we proposed a simple algorithm that iteratively constructs this policy set. Our empirical results (i) validate our theoretical results, (ii) show that the proposed algorithm compares favorably to prior methods, and (iii) demonstrate that a set of independent policies can be useful in scenarios where the downstream task of interest cannot be expressed with a single preference vector, which includes lifelong RL scenarios. Note that our approach relies on the existence of an independent set of features, which are maximized to obtain independent policies. In general, a feature extraction procedure may be needed as a preprocessing step to obtain such features. We hope to tackle this problem in future work. We also hope to find useful generalizations of our theoretical results to MDPs with stochastic transition functions.

ACKNOWLEDGMENTS

This project has been partly funded by an NSERC Discovery grant and the Canada-CIFAR AI Chair program. We would like to thank Shaobo Hou for clarifying some parts of the code and the anonymous reviewers for providing critical and constructive feedback.

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

## A   PROOFS

**Lemma 1.** *Let $\Phi$ be a SIF and let $\pi_i$ be a policy that is induced by the feature $\phi_i \in \Phi$ and is a member of a SIP $\Pi$. Then, the entries of the SF $\psi^{\pi_i}$ of policy $\pi_i$ has the following form:*

$$\psi_j^{\pi_i}(s, a) = \begin{cases} \psi_i^{\pi_i}(s, a), & \text{if } i = j \\ 0, & \text{otherwise} \end{cases}.$$

*Proof.* The proof follows directly from the definition of a SIP. Remember that for a policy $\pi_i$ that is a member of a SIP, we have that:

$$\phi_j(s_t, a_t, s_{t+1}) = \phi_j(s_0, a_0, s_1) \quad \forall j \neq i, \forall i, \forall s_0 \sim d_0 \text{ and } \forall t \in \{1, \dots, T\},$$

where $T$ is the horizon, $a_0 = \pi_i(s_0)$ and $(s_t, a_t, s_{t+1})_{t=1}^T$ is the sequence of state-action-state triples that are generated by $\pi_i$'s interaction with the environment. Thus, for $j \neq i$, we have:

$$\begin{aligned} \psi_j^{\pi_i}(s, a) &= E_{\pi_i, P}\left[ \sum_{k=0}^{\infty} \gamma^k \phi_j(S_{t+k}, A_{t+k}, S_{t+k+1}) \Big| S_t = s, A_t = a \right] \\ &= E_P\left[ \sum_{k=0}^{\infty} \gamma^k \phi_j(S_t, A_t, S_{t+1}) \Big| S_t = s, A_t = a \right] && \text{(by Definition 2)} \\ &= \sum_{k=0}^{\infty} \gamma^k E_P[\phi_j(S_t, A_t, S_{t+1})|S_t = s, A_t = a] \\ &= \sum_{k=0}^{\infty} \gamma^k \phi_j(s, a) \\ &= \sum_{k=0}^{\infty} \gamma^k \phi_j(s_0, a_0) && \text{(by Definition 2)} \\ &= \frac{1}{1-\gamma} \phi_j(s_0, a_0) \\ &= 0. && \text{(by Definition 1)} \end{aligned}$$

$\square$

**Theorem 1.** *Let $\Phi$ be a SIF and let $\Pi$ be a SIP induced by each of the features in $\Phi$. Then, the GPI policy $\pi_\Pi^{\text{GPI}}$ is a solution to the optimization problem defined in (7).*

*Proof.* Remember that the GPI policy for a downstream task $\mathbf{w}$ is obtained as:

$$\pi_\Pi^{\text{GPI}}(s) \in \arg \max_{a \in \mathcal{A}} Q_{r_{\mathbf{w}}}^{\max}(s, a),$$

where $Q_{r_{\mathbf{w}}}^{\max}(s, a) = \max_{\pi_i \in \Pi} Q_{r_{\mathbf{w}}}^{\pi_i}(s, a)$. By expressing $Q_{r_{\mathbf{w}}}^{\pi_i}(s, a)$ as a weighted sum of the SFs of policy $\pi_i$, we have:

$$\begin{aligned} Q_{r_{\mathbf{w}}}^{\max}(s, a) &= \max_i \left[ \sum_{j=1}^{n} w_j \psi_j^{\pi_i}(s, a) \right] \\ &= \max_i \left[ w_i \psi_i^{\pi_i}(s, a) + \sum_{j \neq i} w_j \psi_j^{\pi_i}(s, a) \right] \\ &= \max_i \left[ w_i \psi_i^{\pi_i}(s, a) \right]. && \text{(by Lemma 1)} \end{aligned}$$

Thus, we have:

$$\begin{aligned} \pi_\Pi^{\text{GPI}}(s) &\in \arg \max_{a \in \mathcal{A}} Q_{r_{\mathbf{w}}}^{\max}(s, a) \\ &= \arg \max_{a \in \mathcal{A}} \max_i \left[ w_i \psi_i^{\pi_i}(s, a) \right], \end{aligned}$$

which implies that the GPI policy $\pi_\Pi^{\text{GPI}}$ will obtain all the features associated with a positive $w_i$ (in an order that depends on the product $w_i \psi_i^{\pi_i}(s,a)$), ignore the ones with a zero $w_i$, and avoid the ones with a negative $w_i$. This in turn implies that the GPI policy $\pi_\Pi^{\text{GPI}}$ will solve the optimization problem in (7). □

## B    Experimental Details

In this section, we provide the implementation details of the 2D item collection environment and the experimental details of our policy construction method together with the prior methods that we have used for comparison.

### B.1    Implementation Details of the 2D Item Collection Environment

As described in the supplementary material of Barreto et al. (2020), at each step the agent receives an $11 \times 11 \times (n+1)$ tensor (an $11 \times 11$ image with $(n+1)$ channels) representing the configuration of the environment, where $n$ is the number of items in the environment. The channels are used to identify the items and the walls. Specifically, there is one channel for each of the $n$ items and one channel for the impassable walls around the edges of the grid.

The observations are "egocentric" in the sense that images are shifted so that the agent is always at the top-left cell of the grid. Barreto et al. (2020) found that this representation helps the learning process as each action taken by the agent results in larger changes in the observations. The observations are also toroidal in the sense that the images wrap around the edges so that the agent always observes the environment, even though the walls prevent it from crossing one side to the other.

### B.2    Our Method and DQN

For our experiments with GPE & GPI and DQN (Mnih et al., 2015), we have used the publicly available code[3] of Barreto et al. (2020). Thus, implementation details as the neural network architectures that are used, the hyperparameters (replay buffer sizes, learning rates etc.) and the way the SFs and state-action value functions are learned can all be found both in the provided link and in the supplementary material of Barreto et al. (2020). We have also followed the same experimental protocol provided in the supplementary material of Barreto et al. (2020).

### B.3    Prior Diverse Policy Set Construction Methods

Before moving on to the implementation details of the prior diverse policy set construction methods, we start this section by defining the concept of *reward equivalent policies (REP)*. This definition allows for a simple representation of the policy sets that are constructed by these prior methods in terms of the policy sets that can be induced by the nine preference vectors in Fig. 7.

**Definition 3** (REP). *Let $\pi_i$ and $\pi_j$ be two policies that are induced by tasks $\mathbf{w}_i$ and $\mathbf{w}_j$, respectively. $\pi_i$ and $\pi_j$ are defined to be* REP *if they achieve the same expected total reward on an arbitrary task* $\mathbf{w}$, *i.e. if:*

$$E_{\pi_i,P}\left[\sum_{t=0}^{\infty} r_{\mathbf{w}}(S_t, A_t, S_{t+1})\Big| S_0 \sim d_0\right] = E_{\pi_j,P}\left[\sum_{t=0}^{\infty} r_{\mathbf{w}}(S_t, A_t, S_{t+1})\Big| S_0 \sim d_0\right].$$

It should be noted that even if two policies are induced by two different tasks, they can still be REP. For instance, the policies induced by the tasks (0.6 0.8) and (0.8 0.6) are reward equivalent as they achieve the same expected total reward by collecting all the items in the environment depicted in Fig. 1a. Note that Definition 3 only takes into account what the policies achieve and not how they exactly achieve it.

---

[3]https://github.com/deepmind/deepmind-research/tree/master/option_keyboard

Importantly, Definition 3 allows for defining equivalences between *policy sets*. For instance, the policy set induced by tasks $(0.6\ 0.8)$ and $(0.6\ -0.8)$ contains a policy that collects both of the items (reward equivalent to $\pi_3$) and a policy that collects the first item while avoiding the second one (reward equivalent to $\pi_5$). The policy set $\Pi_{35}$ also contains policies that achieve the same tasks. Thus, the former policy set can be considered to be reward equivalent to the policy set $\Pi_{35}$. Note that another policy set induced by tasks $(0.8\ 0.6)$ and $(0.8\ -0.6)$ can also be considered to be reward equivalent to the policy set $\Pi_{35}$. In the rest of this section, we will use this reward equivalence relation for a simple representation of the policy sets constructed by prior methods (in terms of the policy sets that can be induced by the preference vectors in Fig. 7).

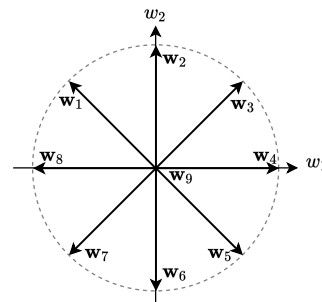

Figure 7: Eight different preference vectors that lie on the surface of the $\ell_2$ 2D ball and a single preference vector ($\mathbf{w}_9$) that is at the origin of this ball. The corresponding values are: $\mathbf{w}_1 = (-\sqrt{1/2}\ +\sqrt{1/2})$, $\mathbf{w}_2 = (0\ +1)$, $\mathbf{w}_3 = (+\sqrt{1/2}\ +\sqrt{1/2})$, $\mathbf{w}_4 = (+1\ 0)$, $\mathbf{w}_5 = (+\sqrt{1/2}\ -\sqrt{1/2})$, $\mathbf{w}_6 = (0\ -1)$ $\mathbf{w}_7 = (-\sqrt{1/2}\ -\sqrt{1/2})$, $\mathbf{w}_8 = (-1\ 0)$, $\mathbf{w}_9 = (0\ 0)$.

**DIAYN** (Eysenbach et al., 2018). Among the prior diverse policy set construction methods that we have used, all except DIAYN are compatible with the GPE & GPI framework. Thus, in order to also make DIAYN compatible, while discovering a set of diverse policies, we have concurrently learned the SFs of the policies in this set so that GPE & GPI can be performed later on. Importantly, rather than encouraging diversity in the state visitation profile, we have encouraged diversity in the *feature visitation profile* by feeding the visited feature vectors $\phi$ as input to the discriminator (as opposed to feeding the visited states as in regular DIAYN). This is because we are interested in policies that are able to obtain certain features, as opposed to ones that just push the agent to different parts of the state space. For the RL algorithm, we have used a vanilla policy gradient algorithm with an additional entropy loss term, for preventing the policy from becoming deterministic in the early stages of training.

After setting the number of policies to be discovered to 3 (as there are 3 possible feature vectors in the environment depicted in Fig. 1a: $(1\ 0)$, $(0\ 1)$ and $(0\ 0)$), learning the SFs of these policies and running the GPE & GPI compatible version of DIAYN described above, we have observed that the discovered policies either end up being policies that obtain both of the items (reward equivalent to $\pi_3$) or policies that obtain none (reward equivalent to $\pi_7$), corresponding to a policy set that is reward equivalent to the policy set $\Pi_{37}$ (see Fig. 7). Thus, the results presented in Fig. 4 corresponds to the results obtained with the policy set $\Pi_{37}$. We have also experimented with larger numbers of policies to be discovered, however, we observed no difference in the qualitative behavior of the policies that were discovered.

In our experiments, for the representation of the policy we have used the same neural network architecture that was provided in Barreto et al. (2020). The hyperparameters of our implementation are provided in Table 1.

Table 1: Hyperparameters of our DIAYN implementation.

| | |
|---|---|
| Learning rate of the policy | $1e-3$ |
| Learning rate of the discriminator | $1e-3$ |
| Discount | 0.95 |
| Entropy coefficient | 0.001 |
| Gradient clip | 1.00 |
| Value function loss coefficient | 0.05 |

**SMP** (Zahavy et al., 2020). After initializing the preference vector to an arbitrary vector on the surface of the $\ell_2$ 2-dimensional ball (see Fig. 7) and running the SMP algorithm till termination, we have observed that for each initialization of preference vector, different policy sets were constructed. The resulting policy sets end up being reward equivalent to one of the following sets: $\Pi_{37}$, $\Pi_{17}$, $\Pi_{157}$, $\Pi_{57}$, $\Pi_{517}$, $\Pi_{27}$, $\Pi_{47}$, $\Pi_7$, $\Pi_{87}$, $\Pi_{857}$, $\Pi_{8517}$, $\Pi_{617}$, $\Pi_{6157}$ and $\Pi_{67}$. Thus, the results presented in Fig. 4 correspond to the average results obtained by these policy sets.

**DSP** (Zahavy et al., 2021). After initializing the preference vector to an arbitrary vector on the surface of the $\ell_2$ 2-dimensional ball (see Fig. 7), setting $T = 2$ (see Algorithm 1 in Zahavy et al. (2021) for the details) and running the DSP algorithm, we have observed that for each initialization of preference vector, different policy sets were constructed. The resulting policy sets end up being reward equivalent to one of the following sets: $\Pi_{37}$, $\Pi_{58}$, $\Pi_{16}$, $\Pi_{27}$, $\Pi_{47}$, $\Pi_{79}$, $\Pi_{68}$ and $\Pi_{86}$. Thus, the results presented in Fig. 4 correspond to the average results obtained by these policy sets.

We have also experimented with larger $T$ values, however, after $T = 2$ the only policy to be added to the set is a policy that is reward equivalent to $\pi_7$. For instance, when $T = 3$, the resulting policy sets end up being reward equivalent to one of the following sets: $\Pi_{377}$, $\Pi_{587}$, $\Pi_{167}$, $\Pi_{277}$, $\Pi_{477}$, $\Pi_{797}$ and $\Pi_{867}$. Since $\pi_7$ is a policy that avoids the items in the environment depicted in Fig. 1a, it is not a useful policy for the tasks considered in this study. Thus, we reported results only for the case of $T = 2$.

### B.4 MaxQInit

MaxQInit (Abel et al., 2018) is a lifelong RL approach in which the value function of an agent is initialized to the best possible value that minimizes the learning time in a new downstream tasks, while at the same time preserving PAC guarantees. More specifically, before starting a new task, the value function is initialized as follows:

$$\hat{Q}_{\max}(s, a) = \max_{M \in \hat{\mathcal{M}}} Q_M(s, a) \tag{10}$$

where $\hat{\mathcal{M}}$ is the set of MDPs that the agent has sampled so far, and $Q_M$ is the value function that the agent learned from interacting with each MDP. However, as the policy sets ($\Pi_{15}$ and $\Pi_{24}$) generated by Algorithm 1 are obtained as a result of pre-training in the environment, fair a comparison with these policy sets, we have allowed for $\hat{\mathcal{M}}$ to be equal to $\mathcal{M}$, where $\mathcal{M}$ is the set of all MDPs (see (3)).

In the GPE & GPI framework, (10) corresponds to initializing the value functions using to the values of the task $\mathbf{w}_3 = (+\sqrt{1/2} +\sqrt{1/2})$ (see Fig. 7) as it is with this task that $Q_M$ is the maximized. Thus, the results presented in Fig. 6 correspond to the results obtained by initializing the value function of DQN (Mnih et al., 2015) to the value function of the task $\mathbf{w}_3$ right at the beginning of each new task.

## C    Unnormalized Results

In this section, we provide the unnormalized results for the experiments in Section 5. The r- and y-axis of the plots now indicates the sum of rewards as it is, without any normalization.

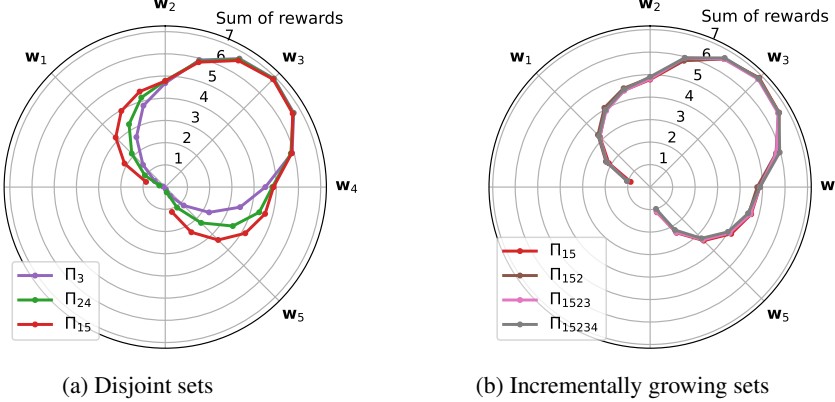

(a) Disjoint sets        (b) Incrementally growing sets

Figure 8: The unnormalized sum of rewards over 17 evenly spread tasks over the nonnegative quadrants of the unit circle. The plots are obtained by averaging over 10 runs with 1000 episodes for each task. The performance comparison of (a) $\Pi_{15}$ with disjoint sets $\Pi_{24}$ and $\Pi_5$, and (b) $\Pi_{15}$ with incrementally growing sets $\Pi_{152}$, $\Pi_{1523}$ and $\Pi_{15234}$.

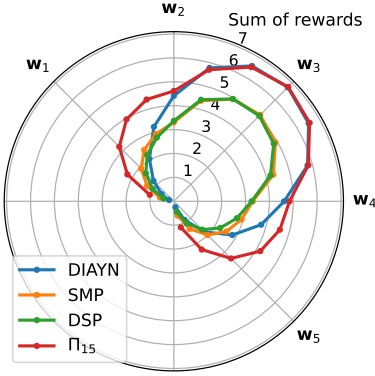

Figure 9: The unnormalized sum of rewards of $\Pi_{15}$, and the policy sets constructed by DIAYN, SMP and DSP. Since the policy sets constructed by the prior methods depend on their particular initialization, their plots are obtained by running each of the constructed policy sets for 5 runs and then averaging over their results. For each task, the agent was evaluated on 1000 episodes. The plot for $\Pi_{15}$ is obtained in a similar way as in Fig. 8.

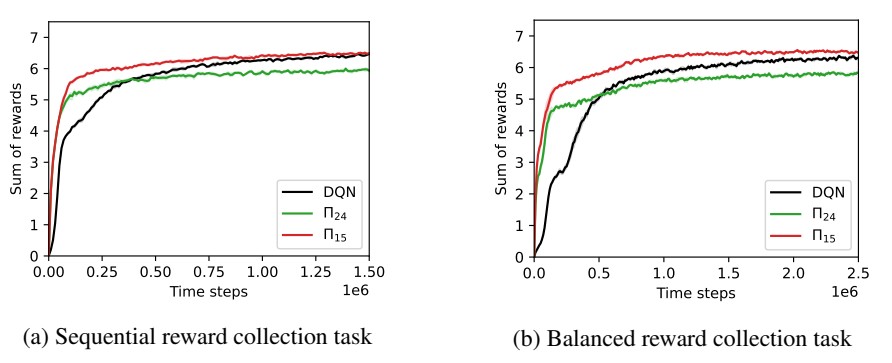

(a) Sequential reward collection task

(b) Balanced reward collection task

Figure 10: The unnormalized sum of rewards of the policy sets $\Pi_{15}$ and $\Pi_{24}$, and DQN on the (a) sequential reward collection and (b) balanced reward collection tasks. Shadowed regions are one standard error over 10 runs.

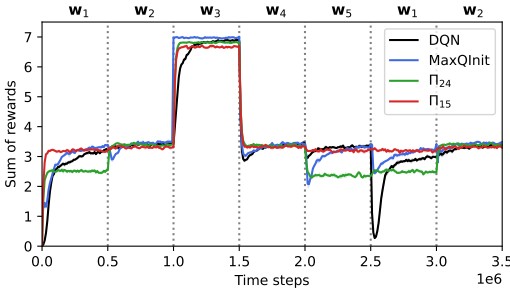

Figure 11: The unnormalized sum of rewards of the policy sets $\Pi_{15}$ and $\Pi_{24}$, DQN, and MaxQInit in a lifelong RL setting described in the text. Shadowed regions are one standard error over 100 runs.

## D   RESULTS FOR STOCHASTIC ENVIRONMENTS

Even though we have developed our theoretical results by assuming MDPs with deterministic transition functions, in order to test the applicability of our results with stochastic transition functions, we also performed experiments in the stochastic version of the 2D item collection environment with "slip" probabilities 0.125 and 0.25. Results are shown in Fig. 12 and 13. As can be seen, even in the stochastic settings, $\pi_{\Pi_{15}}^{\text{GPI}}$ is able to perform better across all downstream tasks compared to $\pi_{\Pi_{24}}^{\text{GPI}}$

and $\pi_{\Pi_3}^{\mathrm{GPI}}$. It can also be seen that adding more policies to the independent policy set $\Pi_{15}$ again has no effect on the downstream task coverage.

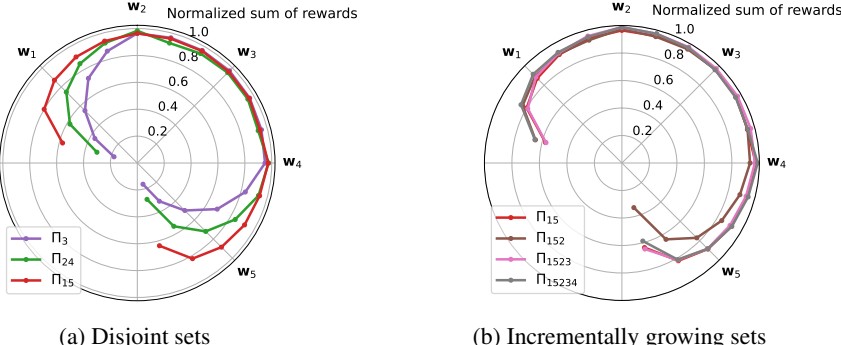

(a) Disjoint sets        (b) Incrementally growing sets

Figure 12: The normalized sum of rewards over 17 evenly spread tasks over the nonnegative quadrants of the unit circle. The results provided in this figure are for the stochastic 2D item collection environment with "slip" probability $0.125$. The plots are obtained by averaging over 10 runs with 1000 episodes for each task. The performance comparison of $\Pi_{15}$ (a) with disjoint sets $\Pi_{24}$ and $\Pi_5$, and (b) with incrementally growing sets $\Pi_{152}$, $\Pi_{1523}$ and $\Pi_{15234}$.

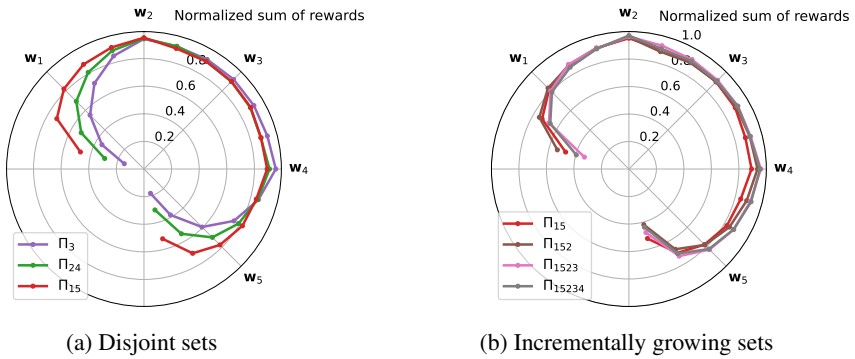

(a) Disjoint sets        (b) Incrementally growing sets

Figure 13: The normalized sum of rewards over 17 evenly spread tasks over the nonnegative quadrants of the unit circle. The results provided in this figure are for the stochastic 2D item collection environment with "slip" probability $0.25$. The plots are obtained by averaging over 10 runs with 1000 episodes for each task. The performance comparison of $\Pi_{15}$ (a) with disjoint sets $\Pi_{24}$ and $\Pi_5$, and (b) with incrementally growing sets $\Pi_{152}$, $\Pi_{1523}$ and $\Pi_{15234}$.

