# OpenReview forum: "Constructing a Good Behavior Basis for Transfer using Generalized Policy Updates"
_ICLR.cc/2022/Conference — ICLR 2022 Poster_

### Official Review · Reviewer_8NxE · 2021-10-26

**Correctness:** 4
**Technical Novelty And Significance:** 2
**Empirical Novelty And Significance:** 2
**Recommendation:** 6
**Confidence:** 3

**Main Review:**

The paper is, in general terms, interesting for the scientific community in RL. It is well structured and written, it reads very well, the introduction correctly states the problem to be addressed (built around previous works), the limitations that we currently find in the literature and, finally, the contributions. I have not found problems in the notation used or in the mathematical formulations.

However, there are a couple of aspects that make me doubt about the merit of this contribution:

It is a clearly incremental work built on the work of Barreto et al. The novelty of the paper and the contributions offered are limited to answering a series of questions (theoretically and experimentally) about the required conditions and the reliability/guarantee provided by the approximations using SFs & GPI.  On the other hand, the experiments shown, although illustrative, are limited and "toy-like". Once again, the settings, frameworks and approximations already presented by Barreto et. al. are used.



**Summary Of The Paper:**

The paper addresses the problem in RL of reusing policies resulting from previously learned (different) tasks in order to perform well on downstream tasks via generalized policy improvement. The authors empirically demonstrate (e.g., using a 2D item collection environment and a lifelong RL scenario) that under certain assumptions about the environment, learning a diverse set of policies bootstrap the learning process on new tasks.

**Summary Of The Review:**

See above.

---

> ### Author Response · Authors · 2021-11-17
> **Response to Reviewer 8NxE**
>
> We would like to thank the reviewer for their feedback on the paper.
>
> A) “It is a clearly incremental work built on the work of Barreto et al. The novelty of the paper and the contributions offered are limited to answering a series of questions (theoretically and experimentally) about the required conditions and the reliability/guarantee provided by the approximations using SFs & GPI.”
>
>   - We are not sure how to effectively address the “clearly incremental” statement of the concern, but we would like to clarify our contributions.
>   - Barreto et al. (2020) [1] proposes the framework of GPE & GPI. However, they leave open two important questions: (1) what set of policies should the agent learn so that its instantaneous performance on **all** possible downstream tasks is guaranteed to be good, and (2) under what conditions does such a set of policies exist.
>   - In this paper, we provide answers to these two questions. It is important to note that answering these questions is very important if we want to tackle lifelong RL using the GPE & GPI framework. For illustrative purposes, let us consider the case where we did not provide answers to these questions. Then, it would not be clear which policies would have to be pre-learned so that the agent is able to solve all possible upcoming tasks. One naive way would be to just add the policies of the encountered tasks to an initially empty policy set of the agent. However, in this case the policy set would grow linearly with the number of tasks that is encountered (which can reach infinity) and it would have no guarantee of downstream task coverage.
>   - In our work, we show (both theoretically and empirically) that just learning $n$ policies (where $n$ is the dimensionality of the features) is enough for maximizing total reward across all possible downstream tasks which is **infinite** in number.
>   - If there is anything that can be done to address the “clearly incremental” concern, we would be interested in knowing about it.
>
> B) “On the other hand, the experiments shown, although illustrative, are limited and "toy-like".”
>
>   - We have performed our experiments in the 2D item collection environment as it is a **prototypical** environment in which the GPE & GPI framework is useful. However, if we performed our experiments in say the 3D item collection environment in Barreto et al. (2018) [2] (which is more complex), then there would be no difference in both our theoretical and empirical results as the ideas would apply straightforwardly. In this case, we would just have to implement a larger neural network with a recurrent module at the end. This is the reason why we have chosen not to perform experiments in additional domains.
>   - We would also like to highlight the fact that despite its 10x10 size, the cardinality of the 2D item collection environment’s state space is $10^{15}$.
>   - Note also that this is the exact same environment that is used by Barreto et al. (2020) [1], whose study we are building on top of.
>   - Another good reason for experimenting with the 2D item collection environment is that it allows for plotting polar plots (see Figure 2 and 4) that visualize the agents performance (total reward) across all possible downstream tasks on the $\ell_2$ 2D ball.
>
> C) “Once again, the settings, frameworks and approximations already presented by Barreto et. al. are used.”
>   - Yes it is indeed true that we use the “setting” and “framework” of Barreto et al. (2020) [1]. However, since we are building on top of their work, we believe that this is natural. If there is anything else we could have done, we would be interested in knowing about it.
>   - Regarding the “approximations”, we are not sure what is meant. However, if this about our theoretical results, we would like to indicate that this is not the case as our theoretical results do not exist in Barreto et al. (2020) [1].
>
> We hope that our response has addressed all the concerns that the reviewer had, but if it didn’t, please point us to the parts that we have missed.
>
> References:
>
> [1] Andre Barreto, Shaobo Hou, Diana Borsa, David Silver, and Doina Precup. Fast reinforcement ´ learning with generalized policy updates. Proceedings of the National Academy of Sciences, 117 (48):30079–30087, 2020.
>
> [2] Andre Barreto, Diana Borsa, John Quan, Tom Schaul, David Silver, Matteo Hessel, Daniel Mankowitz, Augustin Zidek, and Remi Munos. Transfer in deep reinforcement learning using successor features and generalised policy improvement. In International Conference on Machine Learning, pp. 501–510. PMLR, 2018.

---

> > ### Author Response · Authors · 2021-11-26
> > **Response to Reviewer 8NxE**
> >
> > Dear reviewer, with our response above, we have tried to address your concerns regarding our paper. As we are nearing the end of the discussion period, we would like to know if you have any outstanding concerns in light of our response? We will be happy to address them.

---

> > ### Comment · Reviewer_8NxE · 2021-11-29
> > **Response**
> >
> > Thank you very much for the detailed reply. The contributions, although building on previous work, answer new questions theoretically and experimentally. Also, these are now much clearer. All in all, the work is good and interesting, which, together with the excellent work of the authors in the rebuttal phase (with all the justifications and descriptions provided), allows me to increase the mark initially awarded.

---

### Official Review · Reviewer_3QcK · 2021-11-01

**Correctness:** 4
**Technical Novelty And Significance:** 3
**Empirical Novelty And Significance:** 2
**Recommendation:** 6
**Confidence:** 4

**Main Review:**

This paper tackles an interesting and important question that is clearly critical for developing lifelong agents: which policies or value functions should an agent store in its "library" for later use?  The approach taken here is to learn a set of basis policies that are provably optimal (by optimal, we mean that there are no other policies that will result in improved performance after GPI) on downstream tasks within the successor features framework. Given the recent success and adoption of the framework, the results here will likely be of great interest to the community.

I found the paper overall to be extremely well-written and easy to follow (albeit with some details left out, which I will discuss later), and I think that the overall idea is conceptually the correct one concerning the specific question and under the assumptions made. Both theoretical and experimental results are provided, which helps bolster the paper's claims. However, I have certain reservations (and clarifying questions) about some of the results and positioning of the paper, to which I would appreciate a response.

**On related work:**

The big downside for me is the lack of related work, which makes it difficult to determine the novelty and answer the question of why one should pick this method over existing ones. The paper builds on the successor features framework and so, unsurprisingly, there are numerous references and comparisons to this line of prior work.  However, there is insufficient coverage of other related work outside of this, and particularly on approaches that set out to do precisely the same thing studied here.

Of course, the literature regarding the transfer of policies and value functions for multitask RL is large and it is not possible to survey the entire field in the space of a conference paper. However, several approaches accomplish the same kinds of things in this paper and the paper would be made much stronger by distinguishing its contributions from past approaches. For instance, Abel et al (2018) provide theoretical results (and a simple approach that maximises over the value functions) about how best to reuse learned policies to solve new tasks, while van Niekerk et al (2019) study composition in a similar setup (see Theorem 2 in particular). Closely related to this is also Tirinzoni et al (2018).  Fernandez and Veloso (2006) build on Thrun and Schwartz (1995) to determine how to reuse past policies to accelerate learning on new tasks.

The most important piece of related work is the recently published paper by Nemecek and Parr (2021). Their approach tackles the exact same problem, but they are able to iteratively build a policy cache without having to rely on the notion of independent features/policies. One of the most important aspects of their work is that they have motivating experiments as to why you would only want to store certain policies. In the experimental setup here, there are very few tasks, and so why not just store all the policies? In their work, they have 1000+ tasks, and so it is clearly infeasible to store every single policy learned. This motivation is in general missing from the paper. Please comment on the difference between this work and theirs.

It is worth including either direct empirical comparisons, or brief discussions of the differences to this work. Older approaches such as PolicyBlocks (Pickett and Barto 2002) may also be of relevance, since they focus on collecting a cache of policies that are used to inform future learning.

**On theoretical results:**

The theoretical results speak to the idea that, given a set of policies, is a subset we can choose to keep in our "library" to maximise the outcome of doing generalised policy improvement. While this is true, it would have been even nicer to see results that say something about how close to optimal we can get with these policies. For example, closely-related work by Abel et al (2018) provides PAC bounds when maximising a set of previously-learned action value functions, and I can't help but think that a similar kind of result here would be much more impactful.

The definition of independent features seems quite restrictive to me, and from the intuition provided, I can only see it being useful for goal or subgoal-reaching tasks, where the features are 0 everywhere until a subgoal is reached. I'm also not 100% sure I understand the definition of an independent policy. When it says "a policy is induced by each feature", does this mean a policy that "achieves" a feature by reaching a state where its value is non-zero? If so, then I take it to mean that independent policies are policies that cause different features to be set to non-zero (and all others remain 0). Thus, to use the given domain as an example, two independent policies would be a policy that collects triangles, and one that collects diamonds.

If so then I wonder how useful Lemma 1 and the results in Figure 3 are, since the expected features are essentially the Q-value function for solving a task with rewards of 0 everywhere and C when a goal is achieved. These results basically say: the policy for collecting diamonds has a high expected return for collecting diamonds and 0 for collecting triangles (and vice versa), which does not seem particularly surprising.

Furthermore, if this is the case, then the setup seems almost identical to Nangue Tasse et al (2020a) who also introduce the concept of a "policy basis". While that work does not deal with preferences, it seems like it is extremely related, and even if not used as a baseline, it would be helpful to describe how the notion of basis policies differs from that here.

**On empirical results:**

The empirical experiments are well aligned with the theory and the story being told, but I wonder if they do not go far enough. I really liked Figure 2 as an initial start, but there are a few assumptions baked into that result and it would be better if it could be shown how to remove them. In order of importance, these are:
  1. The agent is essentially told which policies are independent, since these weight vectors are provided up front. However, in general the agent does not get to control the order in which it sees the task, and so a better setup would be to have tasks randomly sampled from a distribution, and require the agent to determine whether a policy is independent. This way, the library can be grown in an informed manner, instead of being provided to the agent directly.
  2. The above experiment could also be combined with the regression procedure used in Question 7 to infer the weights. Then the agent would essentially be receiving no information, and would have to learn everything directly from data.
  3. The lifelong experiment doesn't have any "lifelong-oriented" baselines. I think a much stronger experiment would be to test it against MAXQ (Abel et al 2018), Nangue Tasse (2020b) or Nemecek and Parr (2021).
  4. The theory only holds for deterministic cases, but it would be interesting to test empirically in the stochastic setting, which could easily be done by adding a "slip" probability to actions

I also worry a bit about the results in Figure 5. While DQN was used to show what optimal performance looks like, it is remarkably similar to the method presented here. This is despite the fact that DQN must learn everything from scratch, whereas the other approaches only need to learn a mapping from states to weights and then combine them with the pretrained independent (and non-independent) policies. I would have expected the latter approach to completely dominate DQN because of its pretraining, and the fact that this doesn't happen is slightly worrying. Incidentally, the approach taken in Question 6 reminds me of Peng et al (2019) who have a set of policies and attempt to learn weights with which to compose them.

In line with my concern in the theoretical setting, the method here was tested on a single domain that has these nice subgoal properties, but another domain would go a long way to convincing me how applicable the notion of independent features/policies is. What would happen, for instance, if we were in a classic four-rooms domain, and the policies were to get to the doorways and to get to the centre of the rooms? In this case, there's no way to get to the centre of a room without passing through a doorway, and so would things break down here? My main concern again is just how widely applicable these definitions are.

**Clarifying questions and minor comments:**

  1. I realise space is an issue, but it would be better not to direct the reader to a different paper to try find details of the environment setup (which is actually only in the supplementary material of that paper). Please include the description of the domain in the appendix. Incidentally, I could not locate the statement that describes what the feature function is (is it 0 everywhere and 1 when a particular object is picked up?).
  2. What is meant by normalised rewards? Is this just scaling the rewards to be in 0-1? Another thing that could be done here is instead to train optimal value functions, and then show the difference between the returns of the composed policies compared to the optimal returns.
  3. At present, Figure 2 shows that the independent policies are the best, but there's no indication of how far from optimal they are
  4. Typos: "wee" (bottom page 3), missing "to" at top of page 8

**References**

  1. Abel, David, et al. "Policy and value transfer in lifelong reinforcement learning." International Conference on Machine Learning. PMLR, 2018.
  2. Van Niekerk, Benjamin, et al. "Composing value functions in reinforcement learning." International Conference on Machine Learning. PMLR, 2019.
  3. Tirinzoni, Andrea, Rafael Rodríguez-Sánchez, and Marcello Restelli. "Transfer of Value Functions via Variational Methods." NeurIPS. 2018.
  4. Fernández, Fernando, and Manuela Veloso. "Probabilistic policy reuse in a reinforcement learning agent." Proceedings of the fifth international joint conference on Autonomous agents and multiagent systems. 2006.
  5. Thrun, Sebastian, and Anton Schwartz. "Finding structure in reinforcement learning." Advances in neural information processing systems (1995): 385-392.
  6. Nemecek, Mark W., and Ron Parr. "Policy Caches with Successor Features." International Conference on Machine Learning. PMLR, 2021.
  7. Pickett, Marc, and Andrew G. Barto. "Policyblocks: An algorithm for creating useful macro-actions in reinforcement learning." ICML. Vol. 19. 2002.
  8. Nangue Tasse, Geraud , Steven James, and Benjamin Rosman. "A Boolean Task Algebra for Reinforcement Learning." Advances in neural information processing systems (2020a).
  9. Nangue Tasse, Geraud , Steven James, and Benjamin Rosman. "Logical Composition in Lifelong Reinforcement Learning." Lifelong Machine Learning Workshop at ICML (2020b).
  10. Peng, Xue Bin, et al. "MCP: Learning Composable Hierarchical Control with Multiplicative Compositional Policies." Advances in Neural Information Processing Systems 32 (2019): 3686-3697.

**Summary Of The Paper:**

The paper extends the successor features framework to answer the following question: which policies should we learn and store so that, when presented with a new task, we achieve the best performance possible? The paper defines the notion of independent policies (forming a kind of basis over policy space), which can then be combined to solve new tasks (whose rewards are expressible as a linear combination of features) immediately. Experimental results support the theoretical results and show that it is best to learn these independent policies if we wish to maximise performance on downstream tasks.

**Summary Of The Review:**

While the ideas presented here are sound and backed by theoretical and empirical results, the lack of context with respect to prior approaches makes it hard to judge the impact and novelty of the work. The experiments, in particular the lifelong learning one, could benefit from more appropriate baselines. Furthermore, I have concerns about whether the assumptions and definitions are too restrictive, reducing the applicability of the approach more generally, especially compared to more recent work (Nemeck and Parr 2021). I am, of course, open to having this score upgraded based on future discussions and changes.

******************************
POST-REBUTTAL
******************************

The paper has been significantly strengthened during the rebuttal period, and as such I am increasing my score. One important point to clarify is the setting - the "unsupervised RL" setting is a bit ambiguous and could lead to confusion, since it could mean the reward-free RL (see Jin et al 2020) and is separate from the lifelong setting. Clarifying this in the paper is important.

The main improvement that can still be made is extending it to the lifelong learning setting, where the agent doesn't get to pick its task/reward function. Such an agent would be fully autonomous and would be able to discover and store a library of policies on the fly. This would be much more in line with existing work, and make comparisons to other work (Abel et al, Nemeck and Parr) apples-to-apples. It would also make Algorithm 1 far more interesting than it currently stands. Nonetheless, I feel there is enough here as it is to vote for acceptance

---

> ### Author Response · Authors · 2021-11-18
> **Response to Reviewer 3QcK**
>
> We would like to thank the reviewer for their very detailed and mostly constructive feedback on the paper.
>
> **0. Some general clarifications regarding the motivation of the paper**:
>
> A) The main motivation of this paper is to answer the two important questions that are left open by Barreto et al. (2020) [1]: (1) what set of policies should the agent learn so that its instantaneous performance on **all** possible downstream tasks is guaranteed to be good, and (2) under what conditions does such a set of policies exist.
>
> B) In addition to answering these questions, we also show how the learned policy set can be useful in more realistic scenarios (Q6 and Q7). Our initial purpose was not to beat any prior relevant work in these relatively realistic settings.
>
> **1. Responses to concerns regarding related work**:
>
> A) “The big downside for me is the lack of related work, which makes it difficult to determine the novelty and answer the question of why one should pick this method over existing ones. The paper builds on the successor features framework and so, unsurprisingly, there are numerous references and comparisons to this line of prior work. However, there is insufficient coverage of other related work outside of this, and particularly on approaches that set out to do precisely the same thing studied here.”
>
>   1. As explained above, the main purpose of this paper is as described in 0A and we have additionally provided 0B.
>   2. That being said, thanks to the reviewers comment, we have realized that we could make our paper stronger by showing how it relates to other work outside the successor features framework and what are its advantages/disadvantages compared to them.
>   3. For this reason, we have **added a whole new subsection** in the “Related Work” section of our paper. This subsection includes 7/10 of the references provided in reviewer's concern above. It also includes a few additional references.
>
> B) “Of course, the literature regarding the transfer of policies and value functions for multitask RL is large and it is not possible to survey the entire field in the space of a conference paper. However, several approaches accomplish the same kinds of things in this paper and the paper would be made much stronger by distinguishing its contributions from past approaches. For instance, Abel et al (2018) provide theoretical results (and a simple approach that maximises over the value functions) about how best to reuse learned policies to solve new tasks, while van Niekerk et al (2019) study composition in a similar setup (see Theorem 2 in particular). Closely related to this is also Tirinzoni et al (2018). Fernandez and Veloso (2006) build on Thrun and Schwartz (1995) to determine how to reuse past policies to accelerate learning on new tasks.”
>
>   1. We have included a discussion on the similarities/differences of our approach and Abel et al. (2018), Niekerk et al. (2019), Fernandez and Veloso (2006) and a couple of more studies (particularly the references provided in the reviewer’s concern) in the new subsection under our “Related Work” section.

---

> > ### Author Response · Authors · 2021-11-18
> > **Response to Reviewer 3QcK cont.**
> >
> > C) “The most important piece of related work is the recently published paper by Nemecek and Parr (2021). Their approach tackles the exact same problem, but they are able to iteratively build a policy cache without having to rely on the notion of independent features/policies. One of the most important aspects of their work is that they have motivating experiments as to why you would only want to store certain policies. In the experimental setup here, there are very few tasks, and so why not just store all the policies? In their work, they have 1000+ tasks, and so it is clearly infeasible to store every single policy learned. This motivation is in general missing from the paper. Please comment on the difference between this work and theirs.
> >
> > It is worth including either direct empirical comparisons, or brief discussions of the differences to this work. Older approaches such as PolicyBlocks (Pickett and Barto 2002) may also be of relevance, since they focus on collecting a cache of policies that are used to inform future learning.”
> >
> >   1. We were not aware of this study by the time of our submission, and we would like to thank the reviewer for pointing this out. In Nemecek and Parr (2021), the agent starts with a policy cache containing policies for each of the base tasks for a given environment and when presented with a new task, calculates the lower and upper bounds at the start state using the current policy cache and compares them. If the size of the gap exceeds their threshold, then the new policy is added to the cache.
> >   2. However, we are not sure how their approach tackles the exact same problem we are tackling. In our study, we are interested in learning a set of policies so that the GPI policy would maximize the total reward across **all** possible downstream tasks, whereas Nemecek and Parr (2021) is not interested in this wide range of performance at all. Instead they start with a base policy set and they try to become better each time an interesting task (according to their threshold) is presented.
> >   3. Another thing to note is that even though we both use the term “base policies”, we mean different things. In our work, the base policies are the set of independent policies such that when GPI applied on top allows for instantaneously achieving maximum total reward in any possible downstream task, whereas in Nemecek and Parr (2021) these policies are not able to achieve this. Rather their base policies are policies that are induced by one-hot task vectors. For instance, for the case in which the features are 2D, their tasks vectors are of the form [1 0] and [0 1], corresponding to the policy set $\Pi_{24}$ in our study.
> >   4. Additionally, they do not rely on the on the notion of independent features/policies, because maximizing total reward across **all** possible downstream tasks is not their major concern.
> >   5. It is also important to note that in the 2D item collection environment there are not a few tasks, but an infinite number of tasks. For instance, the tasks [0.8 0.6] and [0.6 0.8] are two different tasks. In fact, every point in the $\ell_2$ 2D ball constitutes as a different task (through a different $\mathbf{w}$ value).
> >   6. However, if the number of items in the environment is the major concern, then this could easily be increased upto say 1000 items by making the grid significantly larger and none of our results (both theoretical and empirical) would change. However, in this case, we would not be able to use the nice polar plots (Figures 2 and 4) that displays performance across all $\mathbf{w}$ values.
> >   7. Regarding PolicyBlocks (Pickett and Barto, 2002), similar to Nemecek and Parr (2018), it also does not aim to find a set of policies so that the combined policy will maximize the total reward across **all** possible downstream tasks. It also does not make use of the successor features framework which comes with instantaneous transfer guarantees.
> >   8. Lastly, note that we have **included** both Nemecek and Parr (2021) and Pickett and Barto (2002) in our new subsection under the “Related Work” section.

---

> > > ### Author Response · Authors · 2021-11-18
> > > **Response to Reviewer 3QcK cont.**
> > >
> > > **2. Responses to concerns regarding theoretical results**:
> > >
> > > A) “The theoretical results speak to the idea that, given a set of policies, is a subset we can choose to keep in our "library" to maximise the outcome of doing generalised policy improvement. While this is true, it would have been even nicer to see results that say something about how close to optimal we can get with these policies. For example, closely-related work by Abel et al (2018) provides PAC bounds when maximising a set of previously-learned action value functions, and I can't help but think that a similar kind of result here would be much more impactful.”
> > >   1. Yes, it is indeed true that having a bound on the optimality (in terms of the discounted return) of the resultant GPI policy for a given task would be nice. However, a bound of this kind has already been provided by Theorem 2 of Barreto et al. (2017) in which they bound the difference between the value function of the GPI policy and the optimal value function for a given task $\mathbf{w}$. Saying something beyond this (even for the set of independent policies we consider) about the optimality of the GPI policy does not seem to be possible and is out of the scope of this paper. This paper only cares about the GPI policy to solve all possible downstream tasks by maximizing total reward (see Eq. (6)), and not for it to solve them optimally (in the sense of the discounted return). Note that the latter is only possible if the agent has an infinite capacity in which it would store all policies for all of the tasks in its policy set (see the paragraph right after our problem formulation).
> > >   2. Regarding the PAC bounds provided by Abel et al. (2018):
> > >
> > >         i. Abel et al. considers a setting in which the agent learns tasks that are sampled from unknown task distribution and thus having PAC bounds makes sense for quantifying the sample complexity. On the contrary, we consider an unsupervised RL setting (see e.g. [7, 8, 9, 10]) in which the agent can freely (without any sample budget) interact with the environment before getting tested. Because of this, we do not think that PAC bounds are relevant for our work.
> > >
> > >       ii. Another thing to note is that Abel et al. builds on top of the work of R-Max [11] and Delayed Q-Learning [12] which allows for coming up with PAC-MDP guarantees. On the contrary, we build on top of the GPE & GPI framework that comes with no such guarantees.
> > >
> > > B) “The definition of independent features seems quite restrictive to me, and from the intuition provided, I can only see it being useful for goal or subgoal-reaching tasks, where the features are 0 everywhere until a subgoal is reached.”
> > >   1. In the literature “goal-based tasks” are used for tasks in which the episode terminates right after the agent reaches goal states (see Abel et al. (2018) or Nangue Tasse et al. (2020a)).
> > >   2. In this paper we consider a different setting in which the agent does not reach a single goal state during an episode, but can reach multiple ones (picking up multiple items) durings its interaction with the environment and our episodes do not end after this reaching procedure.
> > >   3. However, it should also be noted our setting is the setting that is commonly used with successor features and GPI. For instance, [1, 2, 3, 4, 5] all use tasks of this kind. For this reason, we do not think that the definition of independent features is restrictive.
> > >   4. When the features are not independent however, as pointed out in Sec. 7, we require an orthogonalization procedure that preprocesses the raw features to take the form of independent features. In our current study, we have not detailed this procedure and it is an interesting avenue for future work.
> > >   5. It should also be noted that goal-based tasks are a major concern of Nangue Tasse et al. (2020a), which shares similar motivations with our study.

---

> > > > ### Author Response · Authors · 2021-11-18
> > > > **Response to Reviewer 3QcK cont.**
> > > >
> > > > C) “I'm also not 100% sure I understand the definition of an independent policy. When it says "a policy is induced by each feature", does this mean a policy that "achieves" a feature by reaching a state where its value is non-zero? If so, then I take it to mean that independent policies are policies that cause different features to be set to non-zero (and all others remain 0). Thus, to use the given domain as an example, two independent policies would be a policy that collects triangles, and one that collects diamonds.”
> > > >   1. Yes, by “a policy is induced by each feature”, we mean that a policy “achieves” that feature by reaching a state or a set of states where its value is non-zero.
> > > >   2. And, yes it is indeed true that independent policies are policies that cause different features to be set to non-zero. However, note that these features do not remain non-zero. For instance, taking the 2D item collection environment as an example, consider a policy in the independent policy set that is associated with picking up the triangles. This policy would create a trajectory that would result in the feature related to the triangle objects to be set to 1, from 0, five times during an episode (as there are five triangle objects). And, of course, it would not cause a change in the feature related to the diamond objects.
> > > >   3. Finally, yes the two independent policies in the domain provided would be a policy that collects triangles, and one that collects diamonds.
> > > >
> > > > D) “If so then I wonder how useful Lemma 1 and the results in Figure 3 are, since the expected features are essentially the Q-value function for solving a task with rewards of 0 everywhere and C when a goal is achieved. These results basically say: the policy for collecting diamonds has a high expected return for collecting diamonds and 0 for collecting triangles (and vice versa), which does not seem particularly surprising.”
> > > >   1. The role of Lemma 1 in the paper is to serve the proof of Theorem 1 (see Appendix A) and the role of Figure 3 is to empirically show that the results of Lemma 1 hold.
> > > >   2. Figure 3 also shows how the successor features of a set of independent policies look like compared to a set of policies that is not independent.
> > > >
> > > > E) “Furthermore, if this is the case, then the setup seems almost identical to Nangue Tasse et al (2020a) who also introduce the concept of a "policy basis". While that work does not deal with preferences, it seems like it is extremely related, and even if not used as a baseline, it would be helpful to describe how the notion of basis policies differs from that here.”
> > > >   1. We came across the work of Nangue Tasse et al. (2020a) right after the submission of our work and it is perhaps the most relevant study to our work among the 10 references provided above. We would like to thank the reviewer for providing pointers to it again.
> > > >   2. Nangue Tasse et al. (2020a) and our study both have the common goal of building a minimal policy basis so that performance on all possible downstream tasks are maximized. Our definition of a set of independent policies and their “policy basis” has a similar semantics. However, the are important differences between the two studies:
> > > >
> > > >         i. While they define an entire new framework of Boolean algebra for tasks, that requires extensions to the definitions of the reward and value functions of the environment, we build on top of successor features and GPI which requires no modifications to the environment and is already widely adopted. It is also worth mentioning the theoretical guarantees that the successor features framework provides.
> > > >
> > > >         ii. While their framework assumes an undiscounted setting, the successor features framework assumes a discounted one, which is widely used in RL.
> > > >
> > > >         iii. Their proposed framework only considers goal-based tasks where there are absorbing states (goals) that the agent has to reach, and after reaching there the episode terminates. In addition to being able to handle these tasks (for example when the environment has a single item of each type), our framework also addresses tasks in which the agent has to visit multiple (goal) states during a single episodic interaction. (Note that in the 2D item collection environment, the episodes do not end after picking up a single item.)
> > > >
> > > >   3. One additional thing to note is that, similar to our work, Nangue Tasse et al. (2020a) also considers an unsupervised RL setting where unlimited interaction with the environment is possible. And it also assumes the knowledge of goal-states beforehand (similar to our assumption on the knowledge on tasks in Algorithm 1), which makes it “uninformed” in the sense described in Sec. 3A1 of our response.

---

> > > > > ### Author Response · Authors · 2021-11-18
> > > > > **Response to Reviewer 3QcK cont.**
> > > > >
> > > > > **3. Responses to concerns regarding empirical results**:
> > > > >
> > > > > A) “I really liked Figure 2 as an initial start, but there are a few assumptions baked into that result and it would be better if it could be shown how to remove them. In order of importance, these are:”
> > > > >   1. “1. The agent is essentially told which policies are independent, since these weight vectors are provided up front. However, in general the agent does not get to control the order in which it sees the task, and so a better setup would be to have tasks randomly sampled from a distribution, and require the agent to determine whether a policy is independent. This way, the library can be grown in an informed manner, instead of being provided to the agent directly. 2. The above experiment could also be combined with the regression procedure used in Question 7 to infer the weights. Then the agent would essentially be receiving no information, and would have to learn everything directly from data.”
> > > > >
> > > > >         - Yes, it is true that Algorithm 1 constructs a set of independent policies by learning policies for the tasks in a provided set. And it is also true that in the general setting the agent doesn’t get to control the order in which it sees the tasks during test-time. However, in this study, we assume the **unsupervised RL** setting in which the agent is able to **freely act** in an environment before being tested on downstream tasks. Note that this is a common setting that is used in the literature (see e.g. [7, 8, 9, 10]). It is also assumed in Nangue Tasse et al. (2020a).
> > > > >
> > > > >         - Another thing to note is that even in the proposed setting, in which the tasks are randomly sampled from a distribution, the agent would have to be **provided up-front** with some sort of a detector that detects whether a policy belongs to an independent set or not. For instance, consider a setting in which the agent starts receiving tasks from a randomly sampled distribution and its policy set is initially empty. Using the regression procedure used in Q7, the agent can quickly infer the current $\mathbf{w}$ of the task. However, before adding the policy for this task to its policy set, the agent would have to check the coordinates of the inferred $\mathbf{w}$ to see whether if there is a only a single positive component and whether if all the remaining components are negative. Note that with this, we would just be providing the agent of the notion of independent policies at another level and nothing really changes compared to the unsupervised RL setting we considered.
> > > > >
> > > > >         - It is also important to note that there have been prior approaches in the literature (see [5, 6]) that are “informed” in the sense described in 3Aa and that have tried to construct a policy set that would achieve a good performance across all downstream tasks. However, as can be seen in Q5 (see SMP and DSP in particular), these approaches are not able to construct a policy set that consistently performs well across all downstream tasks.
> > > > >
> > > > >   2. “3. The lifelong experiment doesn't have any "lifelong-oriented" baselines. I think a much stronger experiment would be to test it against MAXQ (Abel et al 2018), Nangue Tasse (2020b) or Nemecek and Parr (2021).”
> > > > >
> > > > >         - We are not sure how this concern is related to Figure 2, thus we assume that it is instead related to Figure 6 and Q7. With this in mind have added a **comparison with MAXQINIT** (Abel et al., 2018) in Figure 6.
> > > > >
> > > > >   3. “4. The theory only holds for deterministic cases, but it would be interesting to test empirically in the stochastic setting, which could easily be done by adding a "slip" probability to actions.”
> > > > >
> > > > >         - We have provided **results on the stochastic versions** of the environment for different slip probabilities. These results can be reached at Appendix D.

---

> > > > > > ### Author Response · Authors · 2021-11-18
> > > > > > **Response to Reviewer 3QcK cont.**
> > > > > >
> > > > > > B) “I also worry a bit about the results in Figure 5. While DQN was used to show what optimal performance looks like, it is remarkably similar to the method presented here. This is despite the fact that DQN must learn everything from scratch, whereas the other approaches only need to learn a mapping from states to weights and then combine them with the pretrained independent (and non-independent) policies. I would have expected the latter approach to completely dominate DQN because of its pretraining, and the fact that this doesn't happen is slightly worrying.”
> > > > > >
> > > > > >   1. We are not sure if the results for DQN are remarkably similar to that of the policy set $\Pi_{15}$ (the set of independent policies). In Figure 5a, DQN reaches the performance of $\Pi_{15}$ only after 1.5 million timesteps, whereas in Figure 5b, it cannot even reach it after 2.5 million timesteps. From this aspect, $\Pi_{15}$ completely dominates DQN.
> > > > > >
> > > > > >   2. On the other hand it is expected from DQN to surpass the non-independent policy set $\Pi_{24}$ after a while (in the figures it passes it after approximately 0.35 and 0.5 million timesteps, respectively) as the GPI policy composed by this set is not optimal (by optimality, we mean optimality according to the measure provided in Eq. (6)).
> > > > > >
> > > > > >   3. However, if instead the relatively quick rise of the DQN curve is the main issue, then this is due to the small size of the considered environment. For instance, if the environment was say 50x50, then the DQN curve would not have been able to rise as quickly as in Figure 5 (we have actually observed this during our initial experimentation with the environment). However, as the role of DQN is just to provide a reference to the maximum achievable performance, we did not report results in larger environments.
> > > > > >
> > > > > > C) “... Incidentally, the approach taken in Question 6 reminds me of Peng et al (2019) who have a set of policies and attempt to learn weights with which to compose them.”
> > > > > >
> > > > > >   1. The approach taken in Q6 is based on the OptionKeyboard [13] framework where skills are combined in the space of cummulants. In contrast. Peng et al. (2019) composes them in the space of policies. [13] provides in its discussion that it is advantageous to do the composition in the space of cumulants because it corresponds to manipulating the goal underlying the skills (see the related work section of [13]).

---

> > > > > > > ### Author Response · Authors · 2021-11-18
> > > > > > > **Response to Reviewer 3QcK cont.**
> > > > > > >
> > > > > > > D) “In line with my concern in the theoretical setting, the method here was tested on a single domain that has these nice subgoal properties, but another domain would go a long way to convincing me how applicable the notion of independent features/policies is. What would happen, for instance, if we were in a classic four-rooms domain, and the policies were to get to the doorways and to get to the centre of the rooms? In this case, there's no way to get to the centre of a room without passing through a doorway, and so would things break down here? My main concern again is just how widely applicable these definitions are.”
> > > > > > >
> > > > > > >   1. We have performed our experiments in the 2D item collection environment as it is a prototypical environment in which the GPE & GPI framework is useful. However, if we performed our experiments in say the 3D item collection environment in Barreto et al. (2018) (which is more complex), then there would be no difference in both our theoretical and empirical results as the ideas would apply straightforwardly. In this case, we would just have to implement a larger neural network with a recurrent module at the end. This is the reason why we have chosen not to perform experiments in additional domains.
> > > > > > >
> > > > > > >   2. In the case of the classic four-rooms domain described above (with eight features corresponding to the doorways and centers of the rooms), if by moving to a goal state (doorways or centers) the agent “achieves” that goal, then it is not possible to have a set of independent policies as the environment **setting does not allow for it**. Thus, Algorithm 1, would not be able to construct a policy set whose GPI policy is optimal (according to the measure in Eq. (6)) across all of the downstream tasks $\mathbf{w}\in \mathcal{W}$. However, that being said, Algorithm 1 will still return a policy set that would best solve the optimization problem provided in Eq. (7), though not as good as in the 2D item collection environment.
> > > > > > >
> > > > > > >   3. It is important to note however that this is **not a limitation** of our work as it is impossible for any policy set construction method to build a policy set whose GPI policy would be able to maximize the total reward across all the tasks $\mathbf{w}\in \mathcal{W}$ in the instantiation of the foor-rooms domain considered in 3Db. For instance, let’s imagine an agent that has an infinite policy set containing solutions to all the tasks $\mathbf{w}\in \mathcal{W}$. Even with its infinite capacity, this agent would **not** be able to solve all the downstream tasks. One task that is obviously impossible to solve in this setting is the task in which the agent has to reach all the centers without passing through the doorways. We would like to highlight again the fact that this is a problem with the proposed environment and not our work.
> > > > > > >
> > > > > > >   4. One way to make **all the tasks solvable** in the foor-rooms domain considered in Sec. 3D2 is to add an additional action which allows the agent to *not* achieve a goal (or not pickup an item) while passing through it. Note that this is exactly what is done in Nangue Tasse et al. (2020a) in which they add a “stay” action that achieves the goal. In this version of the foor-rooms domain, our work will be able to construct, without any problem, a set of independent policies whose GPI policy would solve all possible downstream tasks.
> > > > > > >
> > > > > > > **4. Responses to clarifying questions and minor comments**:
> > > > > > >
> > > > > > > A) “I realise space is an issue, but it would be better not to direct the reader to a different paper to try find details of the environment setup (which is actually only in the supplementary material of that paper). Please include the description of the domain in the appendix. Incidentally, I could not locate the statement that describes what the feature function is (is it 0 everywhere and 1 when a particular object is picked up?).”
> > > > > > >
> > > > > > >   1. We agree that it is better to have a self-contained paper and thus have included the details of the domain in the appendix (see Appendix B.1).
> > > > > > >
> > > > > > >   2. The statement that describes what the feature function is can be reached at the third paragraph of Sec. 5 of our paper. There, we describe it as follows: “... Following Barreto et al. (2020), we define each feature $\phi_i$i as an indicator function signalling whether an item of type $i$ has been picked up by the agent. That is, $\phi_i (s, a, s0 ) = 1$ if taking action $a$ in state $s$ results in picking up an item of type $i$, and $\phi_i (s, a, s0 ) = 0$ otherwise. ...”. The same description can also be reached at the “Defining a Behavior Basis” section of Barreto et al. (2020).

---

> > > > > > > > ### Author Response · Authors · 2021-11-18
> > > > > > > > **Response to Reviewer 3QcK cont.**
> > > > > > > >
> > > > > > > > B) “What is meant by normalised rewards? Is this just scaling the rewards to be in 0-1? Another thing that could be done here is instead to train optimal value functions, and then show the difference between the returns of the composed policies compared to the optimal returns.”
> > > > > > > >
> > > > > > > >   1. Exactly, by normalized rewards, we mean scaling them to be in $[0,1]$. We present the results in this format as it makes the visualization more clear. Otherwise, the reader would have to calculate the maximum rewards for each of the $\mathbf{w}$ values along the $\ell_2$ n-dimensional ball and do the comparison themselves. Note that, previous studies on successor features also present the results in this normalized format (see e.g. [1, 2, 3, 4]). However, if interested, the readers can check the unnormalized results for each of our experiments in Appendix C.
> > > > > > > >
> > > > > > > >   2. Yes, it is also possible to compare the discounted return of the GPI policy with the discounted return of the optimal policy for each of the $\mathbf{w}$ values in the $\ell_2$ n-dimensional ball. However, optimality of the GPI policy in the terms of the discounted return is not the major concern of this paper. Rather, we are interested in maximizing the total reward of the GPI policy on downstream tasks (see the performance measure provided in Eq. (6)). That being said, it follows from Theorem 2 of Barreto et al. (2017) that the set of independent policies provided in this paper is not enough to achieve optimality of the GPI policy in terms of the discounted return, instead an infinite policy set that contains solutions to all the tasks in $\mathbf{w}\in \mathcal{W}$ would be required. Thus we did not include any comparisons that use the discounted return.
> > > > > > > >
> > > > > > > > C) “At present, Figure 2 shows that the independent policies are the best, but there's no indication of how far from optimal they are.”
> > > > > > > >
> > > > > > > >   1. We believe that this concern is related to the concern in 4B. Figure 2 shows that the GPI policy composed using the independent policy set $\Pi_{15}$ is the best in terms of the undiscounted return (total reward) and it shows that the GPI policy composed using this set is indeed optimal w.r.t. the performance measure we provide in Eq. (6). On the other hand, if by “optimality” what is meant is optimality in terms of the discounted return, then yes Figure 2 does not provide any information on this as this was not the major concern of the paper. However, as pointed out in our response 4Bb, the GPI policy composed using the independent policy set $\Pi_{15}$ cannot achieve optimality in terms of the discounted return for the reasons specified again in 4Bb. Thus, we have not chosen to plot how far from optimal they are.
> > > > > > > >
> > > > > > > > We hope that our response has addressed all the concerns that the reviewer had, but if it didn’t, please point us to the parts that we have missed. Finally, we would also like to thank the reviewer for pointing out a few of our typos. We have corrected all of them.

---

> > > > > > > > > ### Author Response · Authors · 2021-11-18
> > > > > > > > > **Response to Reviewer 3QcK cont.**
> > > > > > > > >
> > > > > > > > > References:
> > > > > > > > >
> > > > > > > > > [1] Andre Barreto, Shaobo Hou, Diana Borsa, David Silver, and Doina Precup. Fast reinforcement ´ learning with generalized policy updates. Proceedings of the National Academy of Sciences, 117 (48):30079–30087, 2020.
> > > > > > > > >
> > > > > > > > > [2] Andre Barreto, Will Dabney, Remi Munos, Jonathan J Hunt, Tom Schaul, Hado van Hasselt, and ´ David Silver. Successor features for transfer in reinforcement learning. In Proceedings of the 31st International Conference on Neural Information Processing Systems, pp. 4058–4068, 2017.
> > > > > > > > >
> > > > > > > > > [3] Andre Barreto, Diana Borsa, John Quan, Tom Schaul, David Silver, Matteo Hessel, Daniel Mankowitz, Augustin Zidek, and Remi Munos. Transfer in deep reinforcement learning using successor features and generalised policy improvement. In International Conference on Machine Learning, pp. 501–510. PMLR, 2018.
> > > > > > > > >
> > > > > > > > > [4] Diana Borsa, Andre Barreto, John Quan, Daniel J Mankowitz, Hado van Hasselt, Remi Munos, David Silver, and Tom Schaul. Universal successor features approximators. In International Conference on Learning Representations, 2018.
> > > > > > > > >
> > > > > > > > > [5] Tom Zahavy, Andre Barreto, Daniel J Mankowitz, Shaobo Hou, Brendan O’Donoghue, Iurii Kemaev, and Satinder Singh. Discovering a set of policies for the worst case reward. In International Conference on Learning Representations, 2020.
> > > > > > > > >
> > > > > > > > > [6] Tom Zahavy, Brendan O’Donoghue, Andre Barreto, Volodymyr Mnih, Sebastian Flennerhag, and Satinder Singh. Discovering diverse nearly optimal policies with successor features. arXiv preprint arXiv:2106.00669, 2021.
> > > > > > > > >
> > > > > > > > > [7] Benjamin Eysenbach, Abhishek Gupta, Julian Ibarz, and Sergey Levine. Diversity is all you need: Learning skills without a reward function. In International Conference on Learning Representations, 2018.
> > > > > > > > >
> > > > > > > > > [8] Karol Gregor, Danilo Jimenez Rezende, and Daan Wierstra. Variational intrinsic control. arXiv preprint arXiv:1611.07507, 2016.
> > > > > > > > >
> > > > > > > > > [9] Joshua Achiam, Harrison Edwards, Dario Amodei, and Pieter Abbeel. Variational option discovery algorithms. arXiv preprint arXiv:1807.10299, 2018.
> > > > > > > > >
> > > > > > > > > [10] Steven Hansen, Will Dabney, Andre Barreto, David Warde-Farley, Tom Van de Wiele, and Volodymyr Mnih. Fast task inference with variational intrinsic successor features. In International Conference on Learning Representations, 2019.
> > > > > > > > >
> > > > > > > > > [11] Brafman, R. I. and Tennenholtz, M. R-MAX—a general polynomial time algorithm for near-optimal reinforcement learning. Journal of Machine Learning Research, 3 (Oct):213–231, 2002.
> > > > > > > > >
> > > > > > > > > [12] Strehl, A., Li, L., Wiewiora, E., Langford, J., and Littman, M. PAC model-free reinforcement learning. In Proceedings of the 23rd international conference on Machine learning, pp. 881–888, 2006.
> > > > > > > > >
> > > > > > > > > [13] Andre Barreto, Diana Borsa, Shaobo Hou, Gheorghe Comanici, Eser Aygun, Philippe Hamel, Daniel ¨ Toyama, Jonathan Hunt, Shibl Mourad, David Silver, et al. The option keyboard combining skills in reinforcement learning. In Proceedings of the 33rd International Conference on Neural Information Processing Systems, pp. 13052–13062, 2019.

---

> ### Author Response · Authors · 2021-11-22
> **A Brief Response to Reviewer 3QcK**
>
> Dear reviewer, with this brief response, we would like to summarize the updates (marked in blue) that we have done to our revised version of the paper:
>   - We added a new subsection under the “Related work” section of our paper. We believe that this section further clarifies the contribution of our work compared to prior work. This subsection includes most of the 10 studies that were referenced above. It also includes a few additional references.
>   - In this new subsection, we also explained how our study relates to the study of Nemecek and Parr (2021) [1]. (See the response provided below for a detailed explanation of the differences)
>   - We also added MaxQInit [2] as another baseline to our lifelong RL experiments.
>   - Lastly, we added experiments on the stochastic version of the 2D item collection environment (see Appendix D).
>
> We hope that our updates have addressed all the concerns, but if it didn’t, please point us to the parts that we have missed.
>
> References:
>
> [1] Nemecek, Mark W., and Ron Parr. "Policy Caches with Successor Features." International Conference on Machine Learning. PMLR, 2021.
>
> [2] Abel, David, et al. "Policy and value transfer in lifelong reinforcement learning." International Conference on Machine Learning. PMLR, 2018.

---

> > ### Comment · Reviewer_3QcK · 2021-11-25
> > **Rebuttal response**
> >
> > Dear authors,
> >
> > Thank you for the extremely detailed response - I realise it must have taken a long time to put together, and I appreciate the effort. Thank you for the clarifications and explanations on the variety of points. Please let me know if I've missed any nuance or have a misunderstanding.
> >
> > 1. The new related work section is really nice, and makes it clear where this work sits. Essentially my main takeaway from this new section is that, while constructing and storing a library of policies has been around for a while, this work is about storing ones that allow an agent to best solve any new task within the SF/GPI framework.
> >
> > 2. Regarding Nemecek and Parr, my confusion was because I was under the impression that it was exactly this problem being tackled here, but I see now that it is slightly different. In their work, they are saying: the agent gets tasks and must keep a library of policies, and then decide whether to learn a new policy or solve the task with what it already has. In this work, the idea is: once we can learn this particular set of policies, then we are done --- there is no need to learn any more policies.
> >
> > 3. Regarding Nangue Tasse et al, this line in your response helped clarify the setting you consider here: "unsupervised RL setting where unlimited interaction with the environment is possible". I think the biggest difference, as you say, is that it is not an absorbing goal setting. I do not think that the statement that they "assume(s) the knowledge of goal-states beforehand" is quite correct, since their pseudocode shows that they iteratively construct the goal set as they encounter terminal states. The non-absorbing issue is differentiation enough, so this is really a minor point.
> >
> > 4. I don't quite follow the point in 3A1 (second bullet). It seems like you're suggesting there's an easy way of testing if a policy is independent? I don't quite follow this sentence: "Note that with this, we would just be providing the agent of the notion of independent policies at another level and nothing really changes compared to the unsupervised RL setting we considered". The way I see it, while it is true that in this unsupervised RL setting, nothing changes, wouldn't it mean that the method could be used in the lifelong RL setting (e.g. tasks sampled from some distribution at random and the agent constructs a basis from nothing)? Surely that is a really nice feature of this approach? If so, then that would also make the experiment in Q7 fairer in the sense that the approach here and MAXQINIT both start with nothing.
> >
> > 5. An extremely minor point: from a light reading of the MAXQINIT paper, it seems like there's a slightly different approach they take, where they do optimistic initialisation at first, before they start maximising over action-value functions. This likely doesn't make much of a difference for the baseline itself, but perhaps this could be described in the appendix too.
> >
> > Finally, just for my own understanding. The sentence
> >
> > > Note that our approach relies on the existence of an independent set of features, which are maximized to obtain independent policies. In general, a feature transformation procedure may be needed as a preprocessing step to obtain such features. We hope to tackle this problem in future work.
> >
> > Intuitively, for goal/subgoal-reaching tasks, it feels like there will always be independent features (when the features are the agent achieving those (sub)goals). Is it the case that this approach would likely be applicable in all those settings? And perhaps the cases where it is not applicable are those such as locomotion tasks, where there is no explicit goal to reach?
> >
> > I've updated my score in light of the related work section alone, but I'll need a bit more time to go through the changes made in the new version, and will update my score as soon as I've done that

---

> > > ### Author Response · Authors · 2021-11-26
> > > **Response to Reviewer 3QcK**
> > >
> > > Dear Reviewer,
> > >
> > >   1. “The new related work section is really nice, and makes it clear where this work sits. Essentially my main takeaway from this new section is that, while constructing and storing a library of policies has been around for a while, this work is about storing ones that allow an agent to best solve any new task within the SF/GPI framework.”
> > >         - Yes, this is exactly the case.
> > >
> > >   2. “Regarding Nemecek and Parr, my confusion was because I was under the impression that it was exactly this problem being tackled here, but I see now that it is slightly different. In their work, they are saying: the agent gets tasks and must keep a library of policies, and then decide whether to learn a new policy or solve the task with what it already has. In this work, the idea is: once we can learn this particular set of policies, then we are done --- there is no need to learn any more policies.”
> > >         - Yes, this is exactly one of the differences between our work and Nemecek and Parr’s work.
> > >
> > >   3. “Regarding Nangue Tasse et al, this line in your response helped clarify the setting you consider here: "unsupervised RL setting where unlimited interaction with the environment is possible". I think the biggest difference, as you say, is that it is not an absorbing goal setting. I do not think that the statement that they "assume(s) the knowledge of goal-states beforehand" is quite correct, since their pseudocode shows that they iteratively construct the goal set as they encounter terminal states. The non-absorbing issue is differentiation enough, so this is really a minor point.”
> > >         - Yes, while they consider absorbing goal tasks, our approach can handle both these tasks and the ones in which the agent has to achieve multiple goals within a single fixed-length episode.
> > >         - Regarding the statement that they "assume(s) the knowledge of goal-states beforehand", after taking a look at the paper once again, we realize that this statement is not quite correct as you suggested. And, as explained above, they seem to be constructing a goal set as they encounter goal states. We are sorry for not carefully reading it and causing confusion.
> > >         - Even though this is true, we would like to note that one can imagine that if the goal states are far apart, collecting them might not be that feasible. In this case, Nangue Tasse et al. would probably also have to assume the knowledge of goal-state beforehand.
> > >
> > >   4. “I don't quite follow the point in 3A1 (second bullet). It seems like you're suggesting there's an easy way of testing if a policy is independent? I don't quite follow this sentence: "Note that with this, we would just be providing the agent of the notion of independent policies at another level and nothing really changes compared to the unsupervised RL setting we considered". The way I see it, while it is true that in this unsupervised RL setting, nothing changes, wouldn't it mean that the method could be used in the lifelong RL setting (e.g. tasks sampled from some distribution at random and the agent constructs a basis from nothing)? Surely that is a really nice feature of this approach? If so, then that would also make the experiment in Q7 fairer in the sense that the approach here and MAXQINIT both start with nothing.”
> > >         - Yes, there is an easy way of testing whether a policy is independent or not. By using the regression procedure provided in Q7, we can infer the current task’s $\mathbf{w}$ value and check (just by looking at its components) whether it will induce an independent policy or not.
> > >         - Yes, it is totally possible to learn a set of independent policies in a lifelong RL setting. Let us start by imagining that the agent faces tasks sampled from a given distribution and its policy set is initially empty. During its interaction with a given task it can use the regression procedure in Q7 to infer the current task’s $\mathbf{w}$ value and look at its components to check whether it will induce an independent policy or not. If it does, the agent can add the task’s policy to its set and continue with the other task that comes along the way. By this way, with sufficient enough tasks, the agent would be able to construct an independent policy set and use it for upcoming tasks.
> > >         - So, in addition to the unsupervised RL setting, it is totally possible to build an independent policy set in the lifelong RL setting.
> > >         - And, yes, we agree that it would be very nice if we had considered the setting where the agent adds policies to its initially empty set while interacting with the environment. However, as stated before, in this study, we are interested in the unsupervised RL setting and thus we conducted experiments only on this setting.

---

> > > > ### Author Response · Authors · 2021-11-26
> > > > **Response to Reviewer 3QcK cont.**
> > > >
> > > >   4. cont.
> > > >         - Regarding the fairness of MaxQInit, to make a fair comparison with MaxQInit, we have assumed that MaxQInit also has access to **all** the possible MDPs before we ran it in Q7 (the details can be found in Appendix B.4.). So, in a sense, we have brought MaxQInit to the unsupervised RL setting.
> > > >
> > > >   5. “An extremely minor point: from a light reading of the MAXQINIT paper, it seems like there's a slightly different approach they take, where they do optimistic initialisation at first, before they start maximising over action-value functions. This likely doesn't make much of a difference for the baseline itself, but perhaps this could be described in the appendix too.”
> > > >         - We are not sure that we understand this concern. In our implementation of MaxQInit we are also doing an optimistic initialization and then we are maximizing performance on the current task. Note that we are reinitializing MaxQInit to the optimistic value possible at the start of every new task in the task sequence $\mathbf{w}_1, \mathbf{w}_2, \dots$. More on the implementation details can be found in Appendix B.4.
> > > >         - We hope that our response has clarified the concern. If it didn’t, we would be happy to further explain the details of our implementation.
> > > >
> > > >   6. “Intuitively, for goal/subgoal-reaching tasks, it feels like there will always be independent features (when the features are the agent achieving those (sub)goals). Is it the case that this approach would likely be applicable in all those settings? And perhaps the cases where it is not applicable are those such as locomotion tasks, where there is no explicit goal to reach?”
> > > >         - If by “goal/subgoal-reaching tasks” what is meant is the tasks considered in Nangue Tasse et al., then yes, our approach is for sure applicable in all these settings.
> > > >         - And yes, our approach, at least in its current form, is not directly applicable to locomotion tasks. That’s why we state in the conclusion section that a feature transformation procedure is required as a preprocessing step to obtain independent features.
> > > >
> > > > We hope that our responses have addressed all the concerns above, but if it didn’t, please point us to the parts that we have missed. We would also like to thank the reviewer for an increase in the score.

---

### Official Review · Reviewer_4b5c · 2021-11-03

**Correctness:** 4
**Technical Novelty And Significance:** 4
**Empirical Novelty And Significance:** 4
**Recommendation:** 10
**Confidence:** 4

**Main Review:**

This is a strong paper, methodically and clearly presenting its concepts. It is well written, interesting to read, presenting theory before supporting it with experiments. It addresses the main questions with separate experiments. Based on the theory and results, the presented algorithm is better than prior-art in every case. The experiments beyond the original assumptions show that the new algorithm is valuable and interesting for general RL community -- i.e., the last two experiments (reward cannot be expressed as linear combination of features / lifelong RL) show that it outperforms DQN. It would be interesting to see the algorithm also in another domain than 2D item collection (and especially without the linear-combination assumption), but I deem the scope of the paper already well sufficient.

Typos:
p. 3 - it leaveS open the question
p. 3 - clear by the end OF this section
p. 3 - the problem weE want to tackle
p. 7 - tasks that do noT satisfy
p. 7 - test whether IF learning

**Summary Of The Paper:**

The paper focuses on reinforcement learning problems with known successor features and rewards expressible as their linear combination. Building on recent research, it presents a concept of independent features and independent policies and way to construct them. Theoretically it shows that the set of independent policies and their combination with GPE & GPI is enough to solve any induced task. Experimentally, the authors verify the theory and compare to existing approaches to create policy sets, outperforming all. They also provide a set of relevant questions and answers, supported by separate experiments. Finally, they perform experiments in problems without the linear combination assumption and lifelong RL setting, with positive results.

**Summary Of The Review:**

Well written and interesting paper with strong theory and results, well conducted experiments. Recommending acceptation.

---

> ### Author Response · Authors · 2021-11-17
> **Response to Reviewer 4b5c**
>
> We would like to thank the reviewer for their very positive comments about the paper. The reviewer raised a minor concern about the applicability of our approach to other domains than the 2D item collection environment. We have performed our experiments in the 2D item collection environment as it is a prototypical environment in which the GPE & GPI framework is useful. However, if we performed our experiments in say the 3D item collection environment in Barreto et al. (2018) [1] (which is more complex), then there would be no difference in both our theoretical and empirical results as the ideas would apply straightforwardly. In this case, we would just have to implement a larger neural network with a recurrent module at the end. This is the reason why we have chosen not to perform experiments in additional domains. However, experimenting with more complex environments would be an interesting avenue for future work.
>
> We hope that our response has addressed the minor concern that the reviewer had. Finally, we would also like to thank the reviewer for pointing out our typos. We have corrected all of them.
>
> References:
>
> [1] Andre Barreto, Diana Borsa, John Quan, Tom Schaul, David Silver, Matteo Hessel, Daniel Mankowitz, Augustin Zidek, and Remi Munos. Transfer in deep reinforcement learning using successor features and generalised policy improvement. In International Conference on Machine Learning, pp. 501–510. PMLR, 2018.

---

> > ### Comment · Reviewer_4b5c · 2021-11-27
> > **After discussion**
> >
> > Dear authors,
> >
> > thank you for the response. I was watching closely the discussion with the reviewer 3QcK who raised several important concerns and it seems that most of them are satisfactorily resolved now.
> >
> > After the discussion, I still think that the paper is a clear accept. Moreover, given also the technical quality and presentation, I also recommend highlighting on the conference. Hence, I think the original score of 10 is appropriate.
> >
> > Good luck.

---

### Author Response · Authors · 2021-11-18
**Revised version**

Dear reviewers, we have uploaded a revised version of our paper to reflect your comments. For your convenience, most of the changes are marked in blue color.

---

### Decision · Program_Chairs · 2022-01-20

**Decision:**

Accept (Poster)

**Comment:**

This work extends the successor feature framework by focusing on the question of which policies should be learned in order to get the best generalization performance. The reviewers all agree that the question being addressed is interesting and important. One concern raised by two of the reviewers is that the work is rather incremental, providing a relatively small extension from the work of Barreto et al. Nevertheless, the authors have provided a convincing rebuttal, resulting in an increase in score of two of the reviewers. Hence, I recommend acceptance. I do want to ask the authors to carefully read the post-rebuttal point mentioned by reviewer 3QcK about clarifying the unsupervised RL setting.